# Progressive Tempering Sampler with Diffusion

**Severi Rissanen** [* 1]  **RuiKang OuYang** [* 2]  **Jiajun He** [2]  **Wenlin Chen** [2 3]  **Markus Heinonen** [1]  **Arno Solin** [1]
**José Miguel Hernández-Lobato** [2]

## Abstract

Recent research has focused on designing neural samplers that amortize the process of sampling from unnormalized densities. However, despite significant advancements, they still fall short of the state-of-the-art MCMC approach, Parallel Tempering (PT), when it comes to the efficiency of target evaluations. On the other hand, unlike a well-trained neural sampler, PT yields only dependent samples and needs to be rerun—at considerable computational cost—whenever new samples are required. To address these weaknesses, we propose the Progressive Tempering Sampler with Diffusion (PTSD), which trains diffusion models sequentially across temperatures, leveraging the advantages of PT to improve the training of neural samplers. We also introduce a novel method to combine high-temperature diffusion models to generate approximate lower-temperature samples, which are minimally refined using MCMC and used to train the next diffusion model. PTSD enables efficient reuse of sample information across temperature levels while generating well-mixed, uncorrelated samples. Our method significantly improves target evaluation efficiency, outperforming diffusion-based neural samplers.

## 1. Introduction

Sampling from probability distributions is a fundamental problem in many fields of science, including Bayesian inference (Gelman et al., 2013; Welling & Teh, 2011), statistical physics (Von Toussaint, 2011) and molecular simulations

---
[*]Equal contribution  [1]Department of Computer Science, Aalto University, Finland [2]Department of Engineering, University of Cambridge, United Kingdom [3]Department of Empirical Inference, Max Planck Institute for Intelligent Systems, Tübingen, Germany. Correspondence to: Severi Rissanen <severi.rissanen@aalto.fi>, RuiKang OuYang <ro352@cam.ac.uk>, Jiajun He <jh2383@cam.ac.uk>.

*Proceedings of the 42nd International Conference on Machine Learning*, Vancouver, Canada. PMLR 267, 2025. Copyright 2025 by the author(s).

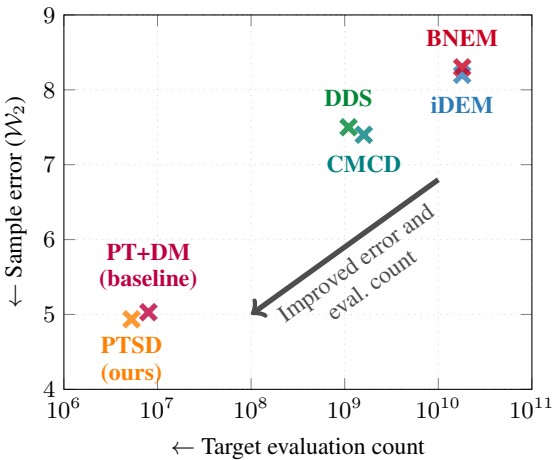

*Figure 1.* Sample error ($\mathcal{W}_2$ distance) and target evaluation times for several diffusion (and control)-based neural samplers on Many-Well-32 target, including DDS, iDEM, BNEM, and CMCD, with our proposed approach. We include the results obtained by first running PT and fit a diffusion model post hoc for comparison.

(Noé et al., 2019). We aim to draw independent samples from a probability distribution with a density:

$$p(x) = \frac{\tilde{p}(x)}{Z} \quad \text{s.t.} \quad Z = \int \tilde{p}(x)\,\mathrm{d}x, \tag{1}$$

where we only assume access to the unnormalized density function $\tilde{p}(x)$ without any ground-truth observations.

The classical way to sample from the target density $p$ is Markov chain Monte Carlo (MCMC), where we design a Markov chain whose invariant density is $p$, with the state-of-the-art method being Parallel Tempering (PT, Swendsen & Wang, 1986; Geyer, 1991; Hukushima & Nemoto, 1996), also known as replica exchange. The key to PT's success lies in running parallel MCMC chains at $K$ temperatures $T_1, \ldots, T_K$, each corresponding to a tempering version $p_{T_k} \propto p^{1/T_k}$ of the original target. The chains at higher temperatures are easier to traverse across the entire support, and samples in those chains are swapped, facilitating better mixing in low temperatures (Woodard et al., 2009). We also note that other choices of path exist, for example, geometric interpolant path $p_k \propto p^{\beta_k} p_0^{1-\beta_k}$ with a tractable reference $p_0$ and $\beta_k$ ranging from 0 to 1, or other

more flexible designs (Syed et al., 2021; Surjanovic et al., 2022). In this paper, we follow the convention in most practical applications to use the annealing path.

However, despite significant advancements in PT, obtaining a new, independent sample requires an independent sample from the highest temperature to be propagated to the lowest temperature (Surjanovic et al., 2024). Consequently, whenever new samples are needed, we need to run PT until uncorrelated samples have been obtained at the highest temperature and transferred to the target, which might require considerable cost. Therefore, a growing trend in research has focused on learned neural samplers, which aim to *amortize* the sampling process and generate uncorrelated samples directly. Early approaches involved fitting normalizing flows to data generated by MCMC (Noé et al., 2019), while more recent efforts typically focus on training generative models directly using only the target unnormalized density. Of particular recent interest are diffusion-based neural samplers, driven by the remarkable success of diffusion models (Ho et al., 2020; Song et al., 2021) in generation tasks. However, a key factor behind the success of diffusion models in these fields lies in their simple and stable denoising score matching (DSM) objective (Vincent, 2011), which relies on access to ground truth data from the target distribution. As a result, while diffusion models demonstrate impressive performance in generation tasks, translating this success to the problem of learning neural samplers directly from unnormalized densities remains challenging.

To illustrate this challenge, in Fig. 1, we compare several recent diffusion and control-based neural samplers, including DDS (Vargas et al., 2023), iDEM (Akhound-Sadegh et al., 2024), BNEM (OuYang et al., 2024), and CMCD (Vargas et al., 2024), on the Many-Well-32 target (Midgley et al., 2023), alongside the results by running PT and fitting a diffusion model post hoc to PT-generated data. Sadly, while showing promising sample qualities, these approaches present significantly lower efficiency compared to directly fitting the diffusion to PT results. This inefficiency arises either from running importance sampling (IS) or annealed importance sampling (AIS) to estimate the objectives (Akhound-Sadegh et al., 2024; OuYang et al., 2024), or from the extensive number of target evaluations in simulating the trajectory (Vargas et al., 2023; 2024) due to the prevalent Langevin preconditioning in the network parameterization (He et al., 2025).

*Does this imply that using PT followed by fitting a diffusion model is the ultimate solution for neural samplers?* We posit that this is not the case. In fact, the methods shown in Fig. 1 represent two ends of the methodological spectrum. On one end, approaches such as DDS, iDEM, BNEM, CMCD, aim to train neural samplers without utilizing any data, while on the other, methods depend exclusively on data generated

through PT and defer model fitting to the final stage.

Therefore, a natural question is to integrate PT and neural samplers—positioned in the middle of the methodological spectrum—leveraging the strengths of both to build more efficient sampling approaches. In this paper, we formalize one attempt towards this direction. Concretely, we propose Progressive Tempering Sampler with Diffusion (PTSD)[1]. Our contributions are as follows:

1. We introduce a novel guidance mechanism that enables diffusion models, trained on higher-temperature data, to generate samples that approximate those from a lower-temperature distribution.

2. Using this guidance term, we propose Progressive Tempering Sampler with Diffusion (PTSD). It fits diffusion models sequentially on higher temperatures and generates samples for lower temperatures using our proposed guidance, allowing the information at higher temperatures to transfer to lower ones efficiently.

3. We evaluate PTSD on a variety of targets, demonstrating its effectiveness. Our approach achieves *orders-of-magnitude* improvement in target density evaluation efficiency compared to other diffusion-based neural samplers, demonstrating a great potential in integrating standard sampling algorithms and neural samplers.

## 2. Background

**Markov chain Monte Carlo and Parallel Tempering** Markov Chain Monte Carlo (MCMC) is a family of algorithms designed to sample from an unnormalized density function by constructing a Markov chain whose stationary distribution matches the target. Representative MCMC methods include Gibbs Sampling (Geman & Geman, 1984), the Metropolis-Hastings (MH) algorithm (Metropolis et al., 1953; Hastings, 1970), the Metropolis-Adjusted Langevin Algorithm (MALA, Grenander & Miller, 1994), and Hamiltonian Monte Carlo (HMC, Duane et al., 1987).

However, for multi-modal distributions, MCMC can easily get stuck in a single mode, failing to explore the entire support effectively in practice (Neal, 1993). Parallel Tempering (PT, Swendsen & Wang, 1986; Geyer, 1991; Hukushima & Nemoto, 1996) addresses this issue by introducing a sequence of temperatures, $T_K > T_{K-1} > \cdots > T_1$, where $T_1$ corresponds to the original target distribution. PT runs multiple Markov chains in parallel, each sampling from a tempered version of the target distribution, defined as $p_{T_k} \propto p^{1/T_k}$, as illustrated in Fig. 2. MCMC at high temperatures can traverse between modes more efficiently, while

---

[1]The code for the paper will be available at https://github.com/cambridge-mlg/Progressive-Tempering-Sampler-with-Diffusion.

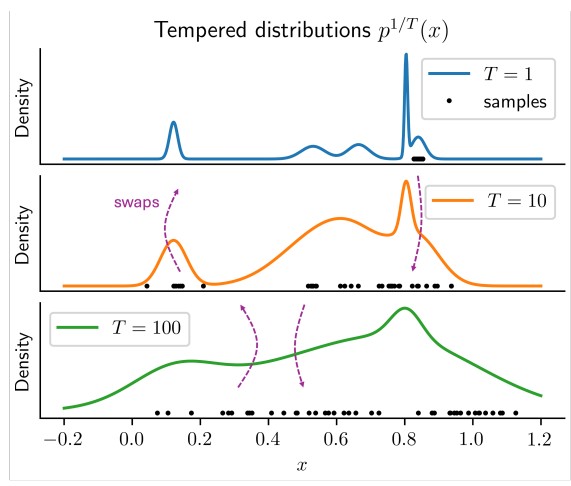

*Figure 2.* Illustration of parallel tempering with three temperatures.

MCMC at lower temperatures explores local modes and provides unbiased samples from the target distribution. Information between different temperatures is shared by swapping samples. Specifically, for samples $x_i$ and $x_j$ at adjacent temperatures $T_i$ and $T_j$, PT swaps them with probability

$$p = \min\left(1, \frac{\tilde{p}(x_j)^{1/T_i}\tilde{p}(x_i)^{1/T_j}}{\tilde{p}(x_i)^{1/T_i}\tilde{p}(x_j)^{1/T_j}}\right). \quad (2)$$

**Diffusion models**   DMs (Sohl-Dickstein et al., 2015; Ho et al., 2020; Song et al., 2021) define a forward process corrupting the original data distribution with a Gaussian noise, which corresponds to a diffusion SDE. We can generate samples by denoising progressively, corresponding to the time-reversal of the forward SDE.

Assume a target density $p_0(x_0)$. We define a noising process[2]

$$dx_t = \sqrt{2\dot{\sigma}(t)\sigma(t)}\,dw_t, \quad x_0 \sim p_0 \quad (3)$$

where $w_t$ is a Wiener process. Its time reversal is given by

$$dx_t = -2\dot{\sigma}(t)\sigma(t)\nabla_{x_t}\log p_t(x_t)\,dt + \sqrt{2\dot{\sigma}(t)\sigma(t)}\,d\bar{w}_t, \quad (4)$$

where $x_{t_{\max}} \sim p_{t_{\max}}$ and $\nabla_{x_t}\log p_t(x_t)$ is the *score function* at diffusion time $t$, and $\bar{w}_t$ is a reverse time Wiener process. The $p_t(x_t)$ is the target density convolved with a Gaussian kernel $p_t(x_t) = \int \mathcal{N}(x_t \,|\, x_0, \sigma(t)^2 I)\,p_0(x_0)\,dx_0$.

For a sufficiently large $\sigma(t_{\max})$, $p_{t_{\max}}$ is well approximated by a Gaussian $\mathcal{N}(x_t \,|\, 0, \sigma(t_{\max})^2 I)$ and hence is tractable. During training, we approximate the score $\nabla_{x_t}\log p_t(x_t)$ with a time-dependent neural network by denoising score

---

[2]For simplicity, we only discuss the variance-exploding process introduced by Song et al. (2021); Karras et al. (2022) in this paper. However, we note that our proposed approach is also compatible with the variance-preserving process.

matching (Vincent, 2011). During sampling, we start with samples from $\mathcal{N}(x_{t_{\max}} \,|\, 0, \sigma(t_{\max})^2 I)$ and follow Eq. (4).

Additionally, the score function is related to the *denoising mean* through Tweedie's formula (Efron, 2011; Roberts & Tweedie, 1996):

$$\nabla_{x_t}\log p_t(x_t) = (\mathbb{E}[x_0 \,|\, x_t] - x_t)/\sigma(t)^2. \quad (5)$$

Therefore, for numerical stability, rather than directly approximating the score function with a neural network, a common choice is to regress the *denoising mean* using a denoiser network $D_\theta(x_t, t)$, as done by Karras et al. (2022).

**PF-ODE and Hutchinson's trace estimator**   Besides the reverse SDE defined in Eq. (4), diffusion models (DMs) can also generate samples by the probabilistic flow (PF) ODE:

$$dx_t = -\dot{\sigma}(t)\sigma(t)\nabla_{x_t}\log p_t(x_t)\,dt, \quad x_{t_{\max}} \sim p_{t_{\max}}. \quad (6)$$

This formulation not only provides an alternative method for sample generation but also offers a principled approach to estimating the log density of the generated samples. Concretely, by employing the instantaneous change-of-variables formula (Chen et al., 2018), we obtain

$$\frac{d\log p_t(x_t)}{dt} = -\dot{\sigma}(t)\sigma(t)\,\text{tr}(\nabla^2_{x_t}\log p_t(x_t)) \quad (7)$$

and the Jacobian can be approximated by Hutchinson's trace estimator (Hutchinson, 1989; Grathwohl et al., 2018)

$$\text{tr}(\nabla^2_{x_t}\log p_t(x_t)) = \mathbb{E}_\epsilon[\epsilon^\top \nabla^2_{x_t}\log p_t(x_t)\epsilon], \quad (8)$$

where $\epsilon$ follows a Rademacher distribution (Hutchinson, 1989). We can approximate the expectation using Monte Carlo integration, with vector-Jacobian products (VJP) enabling efficient computation.

## 3. Methods

In this section, we describe our approach, Progressive Tempering Sampler with Diffusion (PTSD). Unlike other methods shown in Fig. 1, it integrates Parallel Tempering (PT) and neural samplers to achieve more efficient utilization of target energy evaluations.

In spite of the advances by PT, it can only generate independent samples when (1) uncorrected samples are drawn at the highest temperature, and (2) the uncorrected samples are propagated to the lowest temperature. Therefore, the efficiency of PT will highly rely on the quality of local exploration and the swapping. On the other hand, although the MCMC chain can mix faster at higher temperatures, obtaining an uncorrected sample still requires a considerable number of steps. Also, while the (unnormalized) densities at different temperatures differ only in their exponents

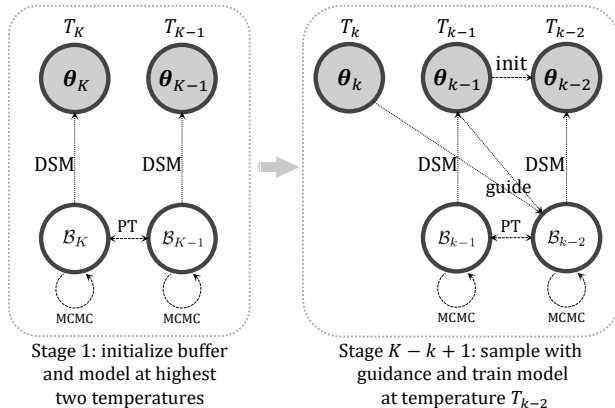

Stage 1: initialize buffer
and model at highest
two temperatures

Stage $K - k + 1$: sample with
guidance and train model
at temperature $T_{k-2}$

*Figure 3.* The training process of PTSD. We unroll the training process into a sequence of target temperatures. We first initialize buffers and models at the highest two temperatures and generate samples from lower temperatures sequentially using the temperature guidance to extrapolate to lower temperatures.

and hence share a global similarity, the swapping is only performed by "copying and pasting" the sample between temperatures, failing to leverage such prior knowledge.

Therefore, a natural question is: *can we find an algorithm to produce uncorrelated samples for higher temperatures, and more efficiently propagate the samples for higher temperatures to lower ones?*

We answer this question affirmatively with a learned neural sampler. A well-trained neural sampler at one temperature can produce independent samples at a cheaper cost. Moreover, the neural sampler can be viewed as a "functional representation" of the target density, and sharing the weights of the neural sampler across temperatures offers a more efficient mechanism for transferring information compared to traditional sample swapping.

With these motivations, we now turn to the details of our approach. We begin by introducing temperature guidance in Sec. 3.1, which is a more efficient way to share information between temperatures by enabling the neural sampler trained at higher temperatures to generate approximate samples for a lower temperature. Then, we integrate this guidance term into the full pipeline of PTSD in Sec. 3.2.

### 3.1. Temperature guidance

To avoid overloading notation, we use uppercase $T$ to denote temperatures while reserving lowercase $t$ for diffusion time steps. Given the target unnormalized density $\tilde{p}$, the unnormalized density function at temperature $T$ is defined by $\tilde{p}^{1/T}$. For simplicity, we will use $p_0(x_0, T)$ to represent the target distribution at temperature $T$, *i.e.* $p_0(x_0, T) \propto \tilde{p}^{1/T}$, and $p_t(x_t, T) = \int \mathcal{N}(x_t \mid x_0, \sigma(t)^2 I) \, p_0(x_0, T) \, dx_0$.

Assume we have a trained diffusion model that approximates

the score function $\nabla_{x_t} \log p_t(x_t, T)$ for $t \in [0, t_{\max}]$ and $T \in [T_1, T_2]$. Now, we want to generate samples at a lower temperature, $T_0 < T_1$, which requires knowing the score function $\nabla_{x_t} \log p_t(x_t, T_0)$. At $t = 0$, this is simple because, by definition, the scores at different temperatures are related by $\nabla_{x_0} \log p_0(x_0, T_0) \cdot T_0 = \nabla_{x_0} \log p_0(x_0, T_1) \cdot T_1$. However, for $t > 0$, this relationship no longer holds in general due to the complex nature of Gaussian convolution.

Instead, we apply Taylor expansion on score around $T_1$:

$$\nabla_{x_t} \log p_t(x_t, T) \approx \nabla_{x_t} \log p_t(x_t, T_1)$$
$$+ (T - T_1) \frac{\partial}{\partial T} \nabla_{x_t} \log p_t(x_t, T) \Big|_{T=T_1}. \quad (9)$$

If the trained model is conditioned on a continuum of temperatures $T$, we could calculate the partial derivative by automatic differentiation. However, this requires training the temperature-conditioned diffusion model on a sufficiently diverse set of temperatures to ensure a robust estimation of the partial derivative.

Fortunately, we can approximate the derivative with finite differences:

$$\frac{\partial}{\partial T} \nabla_{x_t} \log p_t(x_t, T) \Big|_{T=T_1}$$
$$\approx \frac{\nabla_{x_t} \log p_t(x_t, T_2) - \nabla_{x_t} \log p_t(x_t, T_1)}{T_2 - T_1}. \quad (10)$$

Plugging Eq. (10) into Eq. (9), we have

$$\nabla_{x_t} \log p_t(x_t, T) \approx \frac{T_2 - T}{T_2 - T_1} \nabla_{x_t} \log p_t(x_t, T_1)$$
$$- \frac{T_1 - T}{T_2 - T_1} \nabla_{x_t} \log p_t(x_t, T_2) \quad (11)$$
$$= (1 + w) \nabla_{x_t} \log p_t(x_t, T_1) - w \nabla_{x_t} \log p_t(x_t, T_2),$$

where $w = (T_1 - T)/(T_2 - T_1)$. We highlight the similarity between Eq. (11) and the *guidance* in diffusion models (Ho & Salimans, 2021; Karras et al., 2024), offering an intuitive perspective on this estimator: contrasting a "better", lower-temperature model by guiding the model at current temperature $T_1$ with its "worse", higher-temperature version. For this reason, we term Eq. (11) as *temperature guidance*.

One *concern on this guidance* is that when diffusion time $t = 0$, the known relation $\nabla_{x_0} \log p_0(x_0, T) \cdot T = \nabla_{x_0} \log p_0(x_0, T_1) \cdot T_1 = \nabla_{x_0} \log p_0(x_0, T_2) \cdot T_2$ does not hold, indicating this approximate guidance is inaccurate when $t \to 0$. Fortunately, in diffusion models, the accuracy of the score at small time steps typically has a minimal impact on the quality of the generation.

Finally, similar to standard guidance methods, we note that temperature guidance is independent of the specific parameterization of diffusion models. In our experiments, instead of estimating the score function, we follow Karras et al. (2022) to learn the denoising mean using a denoiser.

## 3.2. Progressive Tempering Sampler with Diffusion

We now explore the use of temperature guidance to construct a pipeline that integrates neural samplers with parallel tempering (PT). Analogous to PT, we define a decreasing sequence of temperatures $[T_K, T_{K-1}, \cdots, T_1]$, where $T_1$ corresponds to the temperature of our target distribution.

With the proposed temperature guidance formulation, we structure our algorithm as follows:

1. **Run PT at the highest two temperatures:** To initialize, we run MCMC (specifically, PT with two temperatures) at the two highest temperatures, $T_K$ and $T_{K-1}$, and collect samples into buffers $\mathcal{B}_K$ and $\mathcal{B}_{K-1}$. The Markov chain at these high temperatures is generally easier to explore the support and less likely to get trapped in a local mode (Earl & Deem, 2005).

2. **Fit initial diffusion model:** After obtaining sufficient samples at the two highest temperatures, we fit two diffusion models $\theta_K$ and $\theta_{K-1}$ to each temperature[3].

3. **Draw lower-temperature samples by temperature guidance:** We then draw samples at $T_{K-2}$ using the temperature guidance method proposed in Sec. 3.1 and store these samples in the buffer $\mathcal{B}_{K-2}$.

4. **Fine-tune diffusion model for lower temperature:** We initialize $\theta_{K-2} \leftarrow \theta_{K-1}$, and fine-tune $\theta_{K-1}$ and $\theta_{K-2}$ using samples in buffer $\mathcal{B}_{K-1}$ and $\mathcal{B}_{K-2}$.

We repeat steps 3 and 4 until we obtain diffusion models $\theta_1$.

## 3.3. Improving Techniques for Training PTSD

While the temperature guidance provides an efficient way to guide higher temperature models to sample from lower temperature distributions, we should note that it only generates approximate samples. This approximation error accumulates when running PTSD with multiple temperature levels, ultimately leading to highly biased samples and models. Luckily, we have access to the marginal (unnormalized) density of each temperature at intermediate steps. This enables us to incorporate importance resampling or MCMC steps to refine the quality of the buffer. We now introduce two techniques in detail.

**Truncated importance resampling**  We consider applying importance resampling to intermediate steps at temperature $T_k$. Specifically, we replace the score function in Eq. (6) with the temperature guidance in Eq. (11) and generate samples by following the PF ODE. This allows us to obtain samples $x_1, \ldots, x_B$ along with their corresponding densities $q(x_1), \ldots, q(x_B)$ (in log space). We then compute the

---

[3]From now on, we denote $\theta_k$ as the parameter of the diffusion model trained at temperature $T_k$.

---

**Algorithm 1** Training for PTSD

**Input:** Target density $\tilde{p}$, Temperatures $\{T_k\}_{k=1}^K$, Empty Buffers $\mathcal{B} = \{\mathcal{B}_k\}_{k=1}^K$, Initial parallel tempering (PT) steps $L$, Refinement PT steps $l$, Truncate quantile $\tau$, Training iterations $M$, Buffer size $B$.
**Output:** Model $\theta_1$.
 # Initialize at two highest temperatures:
 Initialize buffers $\mathcal{B}_{K-1}, \mathcal{B}_K$ with $L$ steps PT;
 Train models $\theta_{K-1}, \theta_K$ for $M$ iterations;
 # Progressively decrease the temperature:
 **for** $k$ from $K$ to 3 **do**
  # Sample with temperature-guidance:
  Draw $B$ samples $\{x_n\}_{n=1}^B$ for $T_{k-2}$ by PF ODE with temperature-guidance, using models $\theta_{k-1}, \theta_k$;
  # Calculate Truncated IS Weights:
  Calculate the IS weights $\{w_n\}_{n=1}^B$ by Eq. (12);
  $w_{\max} \leftarrow \tau\text{-}\texttt{quantile}\left(\{w_n\}_{n=1}^B\right)$;
  For $n = 1, \cdots, B$, set $w_n \leftarrow \min(w_n, w_{\max})$;
  Renormalize $\{w_n\}_{n=1}^B$;
  # Importance Resample:
  **for** $i$ from 1 to $B$ **do**
    $n \leftarrow \text{Category}(\{w_n\}_{n=1}^B)$;
    Append $x_n$ to $\mathcal{B}_{k-2}$;
  **end for**
  # Local PT Refinement:
  Refine samples by $l$-step PT in $\mathcal{B}_{k-2}$ and $\mathcal{B}_{k-1}$;
  # Fine-tune models:
  Initialize $\theta_{k-2} \leftarrow \theta_{k-1}$;
  Train $\theta_{k-2}$ on $\mathcal{B}_{k-2}$ for $M$ iterations;
  Train $\theta_{k-1}$ on $\mathcal{B}_{k-1}$ for $M$ iterations;
 **end for**

---

self-normalized importance weights as

$$w_n = \frac{\tilde{p}(x_n)^{1/T_k}/q(x_n)}{\sum_{n'=1}^B \tilde{p}(x_{n'})^{1/T_k}/q(x_{n'})}. \tag{12}$$

Finally, we resample $B$ instances from these $B$ samples with replacement, where the selection probabilities are proportional to $w_n$.

This method is generally guaranteed to be unbiased as $B \to \infty$ and is commonly adopted in Sequential Monte Carlo (SMC, Liu & Chen, 1998). When applying the method with diffusion model proposals, we must be careful, however: the discretization of the PF ODE and the approximation error in Hutchinson's trace estimator (Hutchinson, 1989; Grathwohl et al., 2018) introduce unwanted variance in the normalized importance weights. To prevent the sampling process from becoming unstable, we employ truncated importance sampling (Ionides, 2008), where the unnormalized importance weights are clipped to a maximum value to

reduce variance. In practice, we set the truncation threshold at a predefined quantile of the importance weights.

**Local parallel tempering refinement** While the truncated IS significantly improves sample quality, biases remain due to the approximation error in Hutchinson's trace estimator, the self-normalized importance weights, and the truncation of weights. To address these biases, in addition to IS resampling, we refine the samples by performing several MCMC steps after collecting a buffer. Similar strategies were employed by Sendera et al. (2024); Chen et al. (2024a), where a few steps of MCMC were applied to improve sample quality in buffers.

Additionally, to improve mixing, rather than running MCMC within a single temperature, we can apply PT between pairs of adjacent temperatures. Specifically, assume a buffer size of $B$, after collecting buffer $\mathcal{B}_{k-2}$ at temperature $T_{k-2}$, we randomly pair samples in $\mathcal{B}_{k-2}$ and $\mathcal{B}_{k-1}$ to form $B$ pairs. Then, we run $B$ PT processes in parallel, each initialized with a pair of samples containing two chains at temperatures $T_{k-2}$ and $T_{k-1}$, improving sample quality in both buffers. This approach integrates well with our framework, as we will use both $\mathcal{B}_{k-2}$ and $\mathcal{B}_{k-1}$ to further fine-tune the diffusion models $\theta_{k-2}$ and $\theta_{k-1}$, by which we extrapolate further to $T_{k-3}$.

Optionally, we can optimize energy evaluation usage by only running the PT chains from a subset of IS samples. The results from the PT chains are then added to the original IS results instead of entirely replacing them. Effectively we augment the IS results with the MCMC samples, which can be enough to get around the issue that IS by itself may result in a low effective sample size.

We illustrate the training pipeline in Fig. 3, and detail the pseudo-code in Alg. 1. After training, we sample using the model at the lowest temperature $\theta_1$ by either the reverse-SDE in Eq. (4) or the PF ODE in Eq. (4).

## 4. Connection with Related Works

**Training neural samplers for unnormalized density** There are many approaches aiming to learn a network to sample from the target density without getting access to data. Flow Annealed Importance Sampling Bootstrap (FAB, Midgley et al., 2023) trains a normalizing flow using the $\alpha$-2 divergence, estimated by Annealed Importance Sampling (Neal, 2001), and incorporates a replay buffer (Mnih et al., 2015; Schaul et al., 2016) to reduce computational cost and mitigate forgetting. Recently, due to the success of diffusion models, studies have focused on SDE and control-based neural samplers, such as Path Integral Sampler (PIS, Zhang & Chen, 2022), Denoising Diffusion Samplers (DDS, Vargas et al., 2023), and Controlled Monte Carlo Diffusion (CMCD, Vargas et al., 2024), which match the path mea-

sure between the sampling process and a target process. On the other hand, Iterated Denoising Energy Matching (iDEM, Akhound-Sadegh et al., 2024) estimates the score function directly using the target score identity (De Bortoli et al., 2024) combined with self-normalized importance sampling. Bootstrapped Noised Energy Matching (BNEM, OuYang et al., 2024) generalizes this estimator to energy-parameterized diffusion models, while Diffusive KL (DiKL, He et al., 2024) integrates this estimator with variational score distillation techniques (Poole et al., 2023; Luo et al., 2024) to train a one-step generator as the neural sampler.

A key property of neural samplers is *self-bootstrapping* behavior, as illustrated in Fig. 4. At each training step, the sampler generates samples from itself. Some approaches, such as FAB, iDEM, and BNEM, collect these samples in a buffer. We then use these samples to compute the loss and improve the neural sampler.

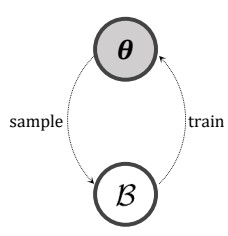

*Figure 4.* "Self-bootstrapping" training of neural samplers.

As the sampler improves, it generates samples that are closer to the target distribution. In turn, these higher-quality samples can provide a stronger training signal, further enhancing the neural sampler in a self-reinforcing cycle.

However, this *self-bootstrapping* behavior can also introduce inefficiencies. The improvement occurs only through the training objective. If the current samples fail to provide a strong enough signal for the objective to refine the neural sampler, the sampler will repeatedly generate similar samples, leading to wasted computation.

In contrast, our approach can be seen as *bootstrapping across temperatures*. We first train models at higher temperatures, where sampling is easier. Then, we use the high-temperature model to generate samples at lower temperatures and leverage these samples to fine-tune the model toward lower temperatures. Additionally, we incorporate importance sampling and PT to refine buffers at each intermediate step. As a result, improvement occurs not only through the training objective but also through the proposed temperature guidance and refinement strategies.

**Integrating MCMC algorithms with neural samplers** Recent line of research has explored integrating MCMC algorithms to improve the performance of neural samplers. They either introduce Sequential Monte Carlo (SMC) to correct bias (Arbel et al., 2021; Albergo & Vanden-Eijnden, 2024; Phillips et al., 2024; Chen et al., 2024a) or employ MCMC to improve the buffer quality (Sendera et al., 2024). However, while these works primarily focus on ensuring

asymptotic correctness or improving the sample quality of neural samplers, they still incur a substantial overhead in evaluating the target energy compared to vanilla PT. In a concurrent work, Zhang et al. (2025) proposed *generalized parallel tempering*, employing a neural network to transport samples between adjacent temperatures and thus boost the swap rate of vanilla PT. In contrast, our work takes a dual perspective: rather than embedding neural samplers into PT to improve PT, we incorporate PT's temperature-exchange mechanism into neural samplers to enhance both sampling efficiency and sample quality of the neural sampler.

**Guidance and inference-time control in diffusion models** Many tools exist for modifying and aggregating pretrained diffusion models at inference time. For instance, it is possible to combine multiple diffusion models to get their intersection (Liu et al., 2022; Du et al., 2023) and add additional constraints and controls to the final distribution (*e.g.*, for inverse problem solving) (Chung et al., 2023; Song et al., 2023; Rissanen et al., 2024). Of particular interest to us are so-called classifier-free guidance (Ho & Salimans, 2021) and its recent development (Karras et al., 2024), where two diffusion model denoisers, one closely fit to the data, and one less closely fit, are contrasted to form diffusion models with even better fits, and hence, better sample quality:

$$D_{\text{guided},w}(x_t) = (1 + w)D_{\theta_1}(x_t) - wD_{\theta_2}(x_t), \quad (13)$$

$D_{\theta_1}$ and $D_{\theta_2}$ represent the well-fitted and poorly-fitted models respectively (Karras et al., 2024). Guidance strength $w > 1$ improves sample quality. When applied to conditional and unconditional models as the 'better' and 'worse' models, the method is called classifier-free guidance (Ho & Salimans, 2021), which has been crucial to the success of text-to-image diffusion models (Saharia et al., 2022; Ramesh et al., 2022; Rombach et al., 2022). This approach closely parallels our temperature guidance method in Eq. (11) and Eq. (13).

## 5. Experiments and Results

In this section, we evaluate our proposed approach and compare with other baselines. In Sec. 5.1, we first test our temperature guidance method on the Lennard-Jones potential with 55 particles (LJ-55), as introduced in (Köhler et al., 2020; Klein et al., 2024). As we will show, this guidance enables effective extrapolation.

In Sec. 5.2, we compare PTSD on two distinct multi-modal distributions, Mixture of 40 Gaussians (MoG-40) and Many-Well-32 (MW-32), with other neural samplers, including FAB (Midgley et al., 2023), iDEM (Akhound-Sadegh et al., 2024), BNEM (OuYang et al., 2024), DiKL (He et al., 2024), DDS (Vargas et al., 2023), and CMCD (Vargas et al., 2024). We also evaluate the performance of a diffusion model trained directly on PT-generated data (PT+DM).

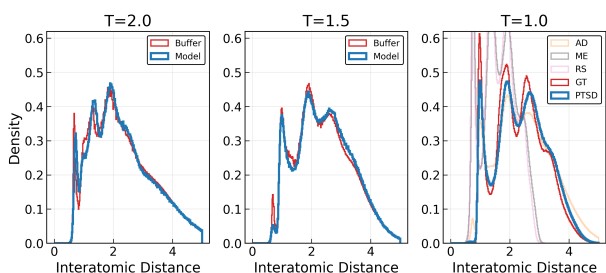

*Figure 5.* First two panels: Marginal density of the interatomic distance on buffers at two temperatures, along with the marginal density of a diffusion model fit to the buffer. Right: The temperature-guided extrapolation based on the higher-temperature models. **AD**: auto-diff extrapolation; **ME**: model extrapolation; **RS**: score rescaling heuristic; **GT**: ground truth; **PTSD**: our temperature guidance.

### 5.1. Extrapolation with Temperature guidance

To showcase the capability of proposed guidance, we conduct experiment on the LJ-55 potential, where we train DMs at temperature 2.0 and 1.5 and aim to extrapolate to a target temperature 1.0. Fig. 5 visualizes the histograms of interatomic distance of samples generated by the trained models at high temperatures, as well as the one of extrapolated samples at the target temperature.

The extrapolation is closer to the target temperature than either model at temperature 2.0 or 1.5, showing a large overlap with the ground truth. We also compare our exploration approach with several other strategies. (1) model extrapolation (ME): as we train a single model on both temperatures 2.0 and 1.5, and by conditioning on the temperature label, we can directly input temperature 1.0 and rely on the generalisation in deep neural networks, an approach that was demonstrated useful for interpolation in Moqvist et al. (2025). (2) auto-diff extrapolation (AD): we directly use Equation (9) instead of finite difference approximation for the time-derivative; (3) score rescaling heuristic (RS): we simply anneal the score with the temperature $\nabla_{x_t} \log p_t(x_t, 1.0) \approx 1.5 \cdot \nabla_{x_t} \log p_t(x_t, 1.5)$, an approach considered by Skreta et al. (2025). Our proposed temperature guidance provides a better extrapolation than the other choices. Even though the extrapolation is not perfect, it serves as an initialization for the local PT refinement, as proposed in Sec. 3.3.

### 5.2. Comparison with Baselines

We now measure the sample quality of PTSD and compare it with other baselines. We report Wasserstein-2 ($\mathcal{W}_2$) distance, Total Variation distance (TVD), and Maximum Mean Discrepancy (MMD). For all tasks, TVD and MMD are measured over the energy histograms of samples, while $\mathcal{W}_2$ is measured in the samples. The calculation $\mathcal{W}_2$ in LJ-55 task takes the SE(3)-equivariance into account as (OuYang et al.,

*Table 1.* **Comparing PTSD with other neural sampler baselines.** We measure (**best**, second best) the TVD and MMD between Energy histograms, and $\mathcal{W}_2$ distance between data samples. We note that as the energy histograms project the entire data space into one dimension, it can be sensitive to outliers but insensitive to mode coverage. '-' indicates that the method diverges or is significantly worse than others.

| | GMM ($d=2$) | | | MW32 ($d=32$) | | | LJ55 ($d=165$) | | |
|---|---|---|---|---|---|---|---|---|---|
| | TVD ↓ | MMD ↓ | W2 ↓ | TVD ↓ | MMD ↓ | W2 ↓ | TVD ↓ | MMD ↓ | W2 ↓ |
| FAB | $0.23_{\pm 0.01}$ | $0.30_{\pm 0.02}$ | $2.50_{\pm 0.19}$ | $0.32_{\pm 0.01}$ | $0.16_{\pm 0.02}$ | $5.70_{\pm 0.01}$ | - | - | - |
| CMCD | $0.14_{\pm 0.01}$ | $0.08_{\pm 0.01}$ | $3.36_{\pm 0.22}$ | $0.69_{\pm 0.01}$ | $0.62_{\pm 0.01}$ | $7.44_{\pm 0.03}$ | - | - | - |
| DDS | $0.22_{\pm 0.01}$ | $0.10_{\pm 0.03}$ | $4.52_{\pm 0.30}$ | $0.77_{\pm 0.00}$ | $0.70_{\pm 0.02}$ | $7.60_{\pm 0.02}$ | - | - | - |
| iDEM | $0.09_{\pm 0.01}$ | **$0.01_{\pm 0.01}$** | $3.26_{\pm 0.42}$ | $0.87_{\pm 0.01}$ | $0.87_{\pm 0.01}$ | $8.30_{\pm 0.02}$ | - | $0.49_{\pm 0.01}$ | $1.95_{\pm 0.00}$ |
| BNEM | $0.12_{\pm 0.00}$ | $0.07_{\pm 0.01}$ | $2.16_{\pm 0.16}$ | $0.31_{\pm 0.01}$ | $0.05_{\pm 0.01}$ | $8.41_{\pm 0.09}$ | **$0.18_{\pm 0.01}$** | **$0.01_{\pm 0.00}$** | **$1.76_{\pm 0.00}$** |
| DiKL | $0.10_{\pm 0.01}$ | $0.03_{\pm 0.01}$ | $3.23_{\pm 0.24}$ | $0.34_{\pm 0.00}$ | $0.16_{\pm 0.02}$ | $6.59_{\pm 0.10}$ | - | - | - |
| PT+DM | $0.13_{\pm 0.05}$ | $0.06_{\pm 0.06}$ | $3.01_{\pm 0.66}$ | **$0.13_{\pm 0.03}$** | **$0.03_{\pm 0.02}$** | $5.04_{\pm 0.02}$ | $0.64_{\pm 0.02}$ | $0.22_{\pm 0.01}$ | $1.82_{\pm 0.00}$ |
| PTSD | **$0.08_{\pm 0.02}$** | $0.02_{\pm 0.01}$ | **$1.93_{\pm 0.41}$** | $0.14_{\pm 0.05}$ | $0.04_{\pm 0.03}$ | **$4.99_{\pm 0.07}$** | $0.79_{\pm 0.01}$ | $0.19_{\pm 0.00}$ | $1.81_{\pm 0.00}$ |

*Table 2.* Number of target density calls for different approaches to achieve the performance reported in Table 1.

| Algorithm | GMM ($d=2$) | MW32 ($d=32$) | LJ55 ($d=165$) |
|---|---|---|---|
| FAB | $6.6 \times 10^6$ | $7.2 \times 10^9$ | $-$ |
| CMCD | $4.4 \times 10^9$ | $1.6 \times 10^9$ | $-$ |
| DDS | $2.6 \times 10^9$ | $1.1 \times 10^9$ | $-$ |
| iDEM | $5.0 \times 10^{10}$ | $1.8 \times 10^{10}$ | $1.3 \times 10^{10}$ |
| BNEM | $7.5 \times 10^9$ | $1.8 \times 10^{10}$ | $6.4 \times 10^9$ |
| DiKL | $1.2 \times 10^{10}$ | $8.0 \times 10^9$ | $-$ |
| PT+DM | $1.2 \times 10^6$ | $8.0 \times 10^6$ | $1.65 \times 10^5$ |
| PTSD (ours) | **$1.0 \times 10^6$** | **$5.3 \times 10^6$** | **$1.5 \times 10^5$** |

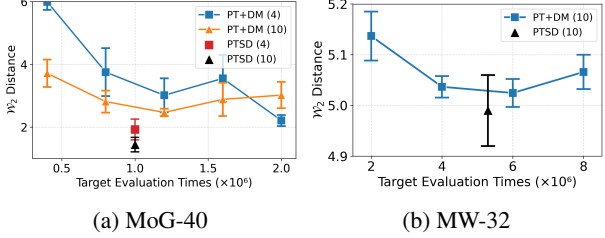

(a) MoG-40         (b) MW-32

*Figure 6.* Comparing PTSD with the results obtained by fitting a diffusion model post hoc to PT-generated data. For MoG-40, we compare two different settings, using 4 or 10 temperature levels.

2024). For LJ-55, we optimize the energy evaluation usage by running PT from a subset of IS samples, as explained in Sec. 3. We note that the energy histogram is sensitive to outliers while generally being robust to mode collapse. In He et al. (2024, Fig. 5), the authors showed that the energy histogram can closely approximate the ground truth, even when significant mode collapse occurs. On the other hand, $\mathcal{W}_2$ distance may provide a more comprehensive assessment of sample quality. We also report the computational cost (in terms of target evaluations) in Table 2.

On both MoG-40 and MW-32, our approach achieves *state-of-the-art sample quality* and demonstrates *orders-of-magnitude improvement in efficiency* over other neural samplers. Our algorithm falls slightly behind on BNEM on sample quality. However, it requires a significantly smaller

number of energy evaluations. In fact, LJ-55 is a relatively simple task for MCMC: a standard MALA requires only 4000 steps to mix. The challenge of LJ-55 for neural samplers arises because the target contains inhibitory regions and has large gradients, leading to instabilities. BNEM addresses this by first smoothing the target density and explicitly regressing the target energy. On the other hand, our approach relies only on samples, making it stable and efficient but less sensitive to inhibitory regions. This explains why we fall slightly behind.

Additionally, to provide a more detailed comparison with PT+DM, we plot the sample quality of PT+DM using samples obtained by PT with different numbers of target evaluations in Fig. 6. As we can see, on these targets, PTSD outperforms PT+DM using the same setting. This gain might come from the fact that PTSD can provide uncorrelated samples for high temperatures easily after being trained on these temperatures, and the temperature guidance serves as an approximate yet more informative "swap" mechanism.

### 5.3. Ablation Study

We conduct ablation studies to verify the effectiveness of temperature guidance and the improvement techniques proposed in Sec. 3.3 in Table 4 on MW-32. As we can see, both the guidance and the truncated IS enhance the performance of our algorithm. Additionally, we note that even without IS, PTSD remains competitive among all neural samplers in Table 1. This further demonstrates the effectiveness of temperature guidance on its own.

### 5.4. Scaling PTSD on Alanine Dipeptide

We also apply the method to the alanine dipeptide molecule in Cartesian coordinates. To maximise the benefits of PTSD, we run local PT refinement only for a subset of IS outputs, as detailed in Sec. 3.3. We evaluate both PTSD and PTDM with approximately the same amount of energy function evaluations ($2.6 \times 10^7$), and show the resulting Ramachandran plots in Fig. 7. This example demonstrates the poten-

*Table 3.* Mean log-likelihood of real data under our PTDM and PTSD models for the alanine dipeptide molecule, and the KL divergence between the ground-truth Ramachandran histogram and the model-generated one from $10^6$ samples.

|  | **PTSD** | **PT+DM** |
|---|---|---|
| $\mathbb{E}_p[\log p_\theta(x)]$ | $\mathbf{213.32 \pm 0.06}$ | $212.38 \pm 0.05$ |
| KLD | 6.9e-2 | **3.2e-02** |

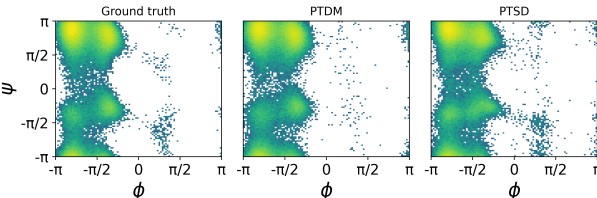

*Figure 7.* Ramachandran plots of ground truth, PT+DM, and PTSD. PT+DM and PTSD were trained with 2.6e7 energy evaluations.

tial scalability and efficiency of PTSD in realistic sampling problems. We also evaluate the log-likelihood of ground-truth data under our PTDM and PTSD models, and the KL divergence between the ground-truth Ramachandran histogram and the model-generated one in Table 3. We show the projection of Ramachandran plots along two axes ($\phi$ and $\psi$) in App. D.4. As we can see, while PT+DM appears slightly better performance on generate the large metastable sates, resulting to be more plausible than PTSD along the $\phi$ axis, PTSD achieves better log-likelihood than PT+DM and also captures the small metastable state more effectively, as shown in Fig. 7. This showcases the great potential of PTSD when applied to more complex systems.

## 6. Conclusions and Limitations

In this paper, we proposed a heuristic temperature guidance that allows us to generate samples at lower temperatures with pretrained diffusion models at two higher temperatures. Based on this, we formulated the Progressive Tempering Sampler with Diffusion (PTSD). PTSD achieved competitive sample quality, demonstrated orders-of-magnitude improvement in efficiency over other neural samplers, and showed promising direction in combining parallel tempering to enhance neural samplers.

While training without data is appealing, this pursuit may be inefficient with current approaches. Instead, designing methods that effectively integrate neural samplers with available data could offer greater practical benefits. While our work may not represent the optimal solution for this direction, it opens a promising avenue for future works, potentially advancing the practicality of neural samplers.

However, several key limitations still remain:

*Table 4.* Ablations on the temperature guidance in Sec. 3.1 and the truncated IS in Sec. 3.3. We do not provide ablation on the local PT refinement as running MCMC is clearly helpful.

|  | TVD ↓ | W2 ↓ |
|---|---|---|
| PTSD w/o temp-guide | 0.34 | 24.59 |
| PTSD w/o IS | 0.23 | 5.84 |
| PTSD | **0.14** | **4.99** |

1. **Network training cost**: while our approach improves efficiency in terms of target evaluations, it remains slower than parallel tempering in terms of wall-clock time in our experiments. This is because we need to fine-tune the diffusion model for each temperature level. For more complex target distributions, we expect that training the diffusion model will be more computationally efficient than directly evaluating the target density and its gradient. Therefore, an important direction for future work is extending our pipeline to more difficult target distributions.

2. **Non-parallelizable execution**: our approach relies on decreasing the temperature progressively, which is different from PT, where different chains can be distributed on multiple devices and run in parallel.

3. **Sensitivity to temperature schedule and network learning quality**: our approach's performance can become fragile when temperature levels differ significantly or when the target distribution is too complex for the network to learn accurately. In contrast, vanilla PT is more robust to a suboptimal temperature schedule and also offers additional flexibility in path selection. The sensitivity also appears in hyperparameters: the network choice, learning rate, and the truncation threshold for truncated IS can also impact the final sample.

Another line of future work is to test the amortizability of our proposed approach. In a recent study, Havens et al. (2025) introduced a conformer-generation dataset comprising many targets with similar properties. Using PT to obtain samples for each target separately and then fitting a diffusion model would be expensive. In contrast, PTSD framework may have the potential to substantially reduce this cost.

Specifically, we could first collect data for each target at the two highest temperatures and fit a diffusion model conditioned on the target label. When performing temperature guidance, we could randomly sample targets to build a buffer with mixed targets. This buffer could be used (with its associated target labels) to train the conditional diffusion model at lower temperatures. Thus, PTSD only requires running separate MCMC during initialization, and it can share information across targets as temperature decreases, resulting in a potential efficiency gain.

## Acknowledgments

We acknowledge Saifuddin Syed for discussions on parallel tempering, and Yuanqi Du, Mingtian Zhang, Louis Grenioux, Laurence Midgley and Javier Antorán for discussions on neural samplers. JH acknowledges support from the University of Cambridge Harding Distinguished Postgraduate Scholars Programme. JMHL and RKOY acknowledge the support of a Turing AI Fellowship under grant EP/V023756/1. SR, MH, and AS acknowledge funding from the Research Council of Finland (grants 339730, 362408, 334600). This project acknowledges the resources provided by the Cambridge Service for Data-Driven Discovery (CSD3) operated by the University of Cambridge Research Computing Service (www.csd3.cam.ac.uk), provided by Dell EMC and Intel using Tier-2 funding from the Engineering and Physical Sciences Research Council (capital grant EP/T022159/1), and DiRAC funding from the Science and Technology Facilities Council (www.dirac.ac.uk). We acknowledge CSC – IT Center for Science, Finland, for awarding this project access to the LUMI supercomputer, owned by the EuroHPC Joint Undertaking, hosted by CSC (Finland) and the LUMI consortium through CSC. We acknowledge the computational resources provided by the Aalto Science-IT project.

## Impact Statement

This paper presents work whose goal is to advance the field of Machine Learning. There are many potential societal consequences of our work, none of which we feel must be specifically highlighted here.

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

# Appendices

## A. Target Distributions

**Mixture of 40 Gaussians (MoG-40)** is a mixture of Gaussians with 40 components in 2-dimensional space, proposed by Midgley et al. (2023). Each component of the MoG-40 has a mean within $[-40, 40] \times [-40, 40]$ and an diagonal covariance of `softplus(1)`.

**Many-Well-32 (MW-32)** is a multi-modal distribution in 32-dimensional space with $2^{32}$ modes, proposed by Midgley et al. (2023). It is constructed by concatenating 16 independent samples from the Double-Well distribution (DW-2, Noé et al., 2019; Wu et al., 2020) in 2-dimensional space. While sampling 16 samples independently from DW-2 and stacking them to construct a MW-32 sample is easy, directly drawing samples from the MW-32 distribution is challenging as a consequence of its multi-modal nature.

**Lennard-Jones-$n$ (LJ-$n$)** describes a $n$-particle system, where the energy between two particles is described by the Lennard-Jones potential and the system energy is given by the sum of pairwise 2-particle energy, *i.e.*

$$V(r) = 4\epsilon \left[ \left( \frac{\sigma}{r} \right)^{12} - \left( \frac{\sigma}{r} \right)^6 \right] \quad \text{and} \quad U(X) = \sum_{i \in [n]} \sum_{i > i} V(r_{ij}), \tag{14}$$

where $X = (x_1, ..., x_n)$, $r_{ij}$ is the distance between $x_i$ and $x_j$, and $\epsilon$ and $\sigma$ are physical constants. While LJ-$n$ can be challenging in terms of numerical stability since the energy of system can be problematically large when particles are too close to each others, it remains a relatively simple target for MCMC with proper initialization or numerically stable implementation. Therefore, we only employ LJ-55 for showcasing the capability of our proposed extrapolation. To ensure numerical stability, we employ cubic spline interpolation introduced by Moore et al. (2024) to smooth the extreme energy values when two particles are close to each other. In our main experiment, we consider LJ-55. While for the ablation study, we conduct experiments on both LJ-13 and LJ-55.

**Alanine Dipeptide (ALDP)** is a 22-particle system formed by two Alanine amino acids. We consider an implicit solvent with a temperature of $300K$, which is used by Midgley et al. (2023). We use the implementation in Midgley et al. (2023) to calculate the energy. In contrast to FAB, which generates samples in the internal coordinate system, we consider generating samples in the Cartesian coordinate system.

## B. Evaluation Metrics

**Wasserstein-2 distance ($\mathcal{W}_2$)** measures the difference between two probability distributions in an optimal transport framework. Given empirical samples $\mu$ from the sampler and ground truth samples $\nu$, the $\mathcal{W}_2$ distance is defined as:

$$\mathcal{W}_2(\mu, \nu) = \left( \inf_{\gamma \in \Pi(\mu,\nu)} \mathbb{E}_\gamma \left[ d^2(X, Y) \right] \right)^{1/2}, \tag{15}$$

where $\Pi$ is the transport plan with marginals distributions $\mu$ and $\nu$ respectively, and $d$ is a distance measure. To calculate the $\mathcal{W}_2$ in practice, we use the Hungarian algorithm implemented in the Python optimal transport package (POT, Flamary et al., 2021), where we employ the Euclidean distance as our distance measure. We measure the $\mathcal{W}_2$ over samples for all tasks. For LJ-55, we calculate the $\mathcal{W}_2$ by taking SE(3), *i.e.* rotation and translation equivariance, into account. In particular, we calculate the distance between two set of samples by $d_{\text{Kabsch}}(X, Y) = \min_{R,t \in \text{SE}(3)} \|X - (YR^\top + t)\|_2$, where Kabsch algorithm is applied to find the optimal rotation and translation.

**Total Variation distance (TVD)** measures the dissimilarity between two probability distributions, which quantifies the absolute differences between two densities over the entire sample space. Given two distribution $P$ and $Q$ defined on a space $\Omega$, with density functions $p$ and $q$, TVD is defined as

$$\text{TVD}(P, Q) = \frac{1}{2} \int_\Omega |p(x) - q(x)| \, dx. \tag{16}$$

We measure the TVD over energy histograms of samples for all tasks.

**Maximum Mean Discrepancy (MMD)** measures the difference between two distributions using functions in a Reproducing Kernel Hilbert Space (RKHS). Given two distributions $P$ and $Q$, the MMD is defined as

$$\text{MMD}(P, Q) = \left( \sup_{f \in \mathcal{H}, \|f\|_{\mathcal{H}} \leq 1} \mathbb{E}_P[f(X)] - \mathbb{E}_Q[f(Y)] \right)^{1/2}, \tag{17}$$

where $f$ is a function in the RKHS associated with a kernel $k(x, y)$ and $\|\cdot\|_{\mathcal{H}}$ a norm defined in this RKHS. The MMD can be computed by using the kernel trick as follows.

$$\text{MMD}(P, Q) = \left( \mathbb{E}_{X,X' \sim P}[k(X, X')] + \mathbb{E}_{Y,Y' \sim Q}[k(Y, Y')] - 2\mathbb{E}_{X \sim P, Y \sim Q}[k(X, Y)] \right)^{1/2}. \tag{18}$$

We measure the MMD over energy histograms of samples for all tasks.

The error intervals in Table 1 for the GMM and MW32 tasks for our method and PTDM were obtained by running the method with multiple seeds, and estimating the standard deviation. Any runs where the metrics calculation resulted in undefined values due to generated outliers in MW32 were discarded. For GMM and MW32 with the baselines and LJ55 in general, we estimated the error by sampling multiple times from a trained model. The error intervals for the log-likelihood evaluations were obtained by estimating the standard deviation of $\log p_\theta(x)$ evaluations and calculating the standard error by dividing by the square root of the amount of samples taken.

## C. Experimental Settings

### C.1. Diffusion Models for PTSD and PT+DM

**Network Architecture.** For non-particle system, *i.e.* GMM-40 and MW-32, we employ a 5-layer MLP. For particle systems, *i.e.* LJ-55 and ALDP, we employ the EGNN implemented by Satorras et al. (2021) with 3 layers.

**Parameterization.** We parameterize the DMs as a denoiser network to approximate the denoising mean following Karras et al. (2022), where the $\sigma_{\text{data}}$ is approximated by the standard deviation of the data used for training, *i.e.* the buffer in PTSD and the drawn samples from PT in PT+DM.

**Noise Schedule.** Our noise schedule follows Karras et al. (2022), with the maximum time $t_{\text{max}} = 40$.

**Sampling Process.** We employ the Euler solver for sampling, where we discretize the time following Karras et al. (2022).

**Parallel Tempering.** Our implementation of parallel tempering is based on the codebase of DiGS (Chen et al., 2024b).

**Energy evaluation optimization.** On both LJ55 and ALDP, we use the method of running PT at the temperature extrapolation steps only from a subset of IS filtered samples, and concatenate the PT samples to the original IS samples. On the GMM and MW32 tasks, we instead run a chain from each diffusion model generated (and resampled) sample, and take only the final output of the chain.

**Temperature levels.** For all experiments, we use a range of geometrically spaced temperatures from a maximum temperature to the desired minimum temperature.

### C.2. Hyperparameters

**PTSD.** Hyperparameter settings for PTSD on each task is illustrated in 5. The key hyperparameters are *the temperature range*, *number of temperatures*, *buffer size* (amount of samples to train the diffusion models on), *the number of initial PT steps at the highest two temperatures*, *the number of steps in PT while following the extrapolation steps*, and the *number of PT chains to use at the extrapolation step*. For MW32, we did not perform the importance resampling at the last extrapolation step.

**PT+DM.** Hyperparameter settings for running PT on different targets are illustrated in Table 6. The *batch size* for DM training is the same as PTSD, and we train the DM until converged.

*Table 5.* Hyperparameter settings for PTSD on different targets.

| Hyperparameters↓ Target→ | MoG-40 | MW-32 | LJ-55 | LJ-55 (illustrative) |
|---|---|---|---|---|
| Temperature range | $[1, 100]$ | $[1, 10]$ | $[1, 3]$ | $[1, 2]$ |
| Number of temperatures | 10 | 10 | 3 | 3 |
| Temperature schedule | geom | geom | geom | linear |
| Buffer Size | 10000 | 12000 | 20000 | 20000 |
| Batch size | 1000 | 1000 | 1000 | 1000 |
| Number of initial PT chains | 100 | 20 | 1 | 1 |
| Number of initial PT steps | 1000 | 20000 | 40000 | 40000 |
| PT swap interval | 5 | 5 | 5 | 5 |
| Burn-in at the initial PT | 100 | 10000 | 5000 | 5000 |
| Interval for subsampling the initial PT chain | 9 | 10 | 1 | 1 |
| Number of generated samples at extrapolation | 10000 | 12000 | 20000 | 20000 |
| Number of PT chains at extrapolation | 10000 | 12000 | 10 | 10 |
| Number of PT steps after extrapolation | 5 | 25 | 2500 | 2500 |
| Number of training iterations | 10000 | 10000 | 120000 | 120000 |
| Importance resampling at last step | Yes | No | Yes | Yes |

*Table 6.* Hyperparameter settings for PT on different targets.

| hyperparams↓ target→ | MoG-40 | MW-32 | LJ-55 | LJ-55 (illustrative) |
|---|---|---|---|---|
| Temperature range | $[1, 200]$ | $[1, 10]$ | $[1, 3]$ | $[1, 2]$ |
| Num. of temperatures | 4 | 10 | 3 | 3 |
| Temperature schedule | geom | geom | geom | linear |

**FAB.** For both MoG-40 and WM-32, we run FAB following exactly the setting by Midgley et al. (2023), with the code at `https://github.com/lollcat/fab-torch/`. As they use buffer which can influence the target density evaluation time, we directly count the time when running the code.

**DDS.** We evaluate DDS using the implementation by Blessing et al. (2024) with KL divergence following Vargas et al. (2023). For MoG-40, we train DDS for 10000 iterations with a batch size of 2000, using Euler-Maruyama discretization with 128 steps. Note that DDS's network takes a score term as the input, and hence, the total number of target evaluations is $10000 \times 2000 \times 128$. For WM-32, we apply early stopping at 3200 iterations, and hence, the total number of target evaluations is $3200 \times 2000 \times 128$.

**CMCD.** We evaluate CMCD using the implementation by Blessing et al. (2024) with KL divergence following Vargas et al. (2024). For MoG-40, we train CMCD with a batch size of 2000, using Euler-Maruyama discretization with 128 steps. It takes 17002 iterations to achieve a good performance, and hence the total number of target evaluations is $17002 \times 2000 \times 128$. For WM-32, we train CMCD with a batch size of 2000, using Euler-Maruyama discretization with 256 steps. We found training for more steps may lead to mode collapsing, and hence we early stop at 3200 iterations. The total number of target evaluations is $3200 \times 2000 \times 256$.

**iDEM.** We evaluate iDEM using the implementation by Akhound-Sadegh et al. (2024). For all tasks, we use Eyler-Maruyama discretization with 1000 steps and 100 inner-loops. For MoG-40, we train iDEM for 1000 outer-loops to ensure convergence, with a batch size of 1000. The marginal scores are estimated by 500 MC samples, which is clipped to a maximum norm of 70. The total number of target evaluations is $1000 \times 100 \times 1000 \times 5000$. For MW-32, we train iDEM for 180 outer-loops, increase the number of MC samples to 1000, clip the score to a maximum norm of 1000, and keep the other hyperparameter the same. The total number of target evaluations is $180 \times 100 \times 1000 \times 1000$. For LJ-55, we further clip the score to a maximum norm of 20, decrease the batch size to 128, and train iDEM for 1000 outer-loops. The total number of target evaluations is $1000 \times 100 \times 128 \times 1000$.

**BNEM.** We evaluate BNEM using the implementation by OuYang et al. (2024). For all tasks, we use Eyler-Maruyama discretization with 1000 steps and 100 inner-loops, we also clip the score to a maximum norm of 1000 during sampling. For MoG-40, we train BNEM for 150 outer-loops to ensure convergence, with a batch size of 1000. The noised energies are estimated by 500 MC samples. The total number of target evaluations is $150 \times 100 \times 1000 \times 5000$. For MW-32, we train BNEM with 180 outer-loops and change the number of MC samples to 1000, resulting in the total number of target evaluations to be $180 \times 100 \times 1000 \times 1000$. For LJ-55, we use the almost the same hyperparameters as MW-32, except for decreasing the batch size to 128. We train BNEM for 500 outer-loops. The total number of target evaluations is $500 \times 100 \times 128 \times 1000$.

**DiKL.** We evaluate DiKL with the implementation by He et al. (2024). For Mog-40, we train for 75000 iterations using a batch size of 1024, and we take 15 AIS steps with 10 samples, with an extra MALA of 5 steps. Therefore, the total number of target evaluations is $75000 \times 1025 \times (15 \times 10 + 5)$. Similarly, for MW-32, we train it for 50000 iterations, leading to a total number of target evaluation of $50000 \times 1025 \times (15 \times 10 + 5)$.

### C.3. Training DM for ALDP in Cartesian Space

ALDP are chiral molecules, which exist in two indistinguishable forms (L-form and D-form) that are mirror images of each other. While in nature, the ALDPs are found in L-form only, therefore we are interested in generating L-form samples. However, the EGNN implemented by Satorras et al. (2021) cannot distinguish these two forms, we reflect the D-form samples generated by PT+DM to make them into L-form. For PTSD, since the difference in forms doesn't influence model training, we leave the D-form samples in all buffers while reflecting them at the last stage only for evaluation.

## D. Supplementary Experiments and Results

### D.1. Ablation Study for Extrapolation

In this section, we ablate different methods for extrapolating from higher-temperature DMs to lower-temperature one. In particular, we train two DMs on two temperatures $T_2 < T_3$, then extrapolate to a lower temperature $T_1 < T_2$. We consider the following ways:

1. *NN Generalization*: train a temperature-conditioned DM $D_\theta(x_t, t, T)$ on $T_2$ and $T_3$, then directly generalize to $T_1$ by sampling from $D_\theta(x_t, t, T_1)$.

2. *Auto-diff TE*: train the same temperature-conditioned DM, but extrapolate it through the first-order Taylor expansion by auto-differentiation, *i.e.* Eq. (9).

3. *Finite-difference TE*: train two distinct standard DMs, then extrapolate by Eq. (11).

We conduct experiments on GMM-40, DW-4 [4], LJ-13, and LJ-55 benchmarks. For all experiments, we choose $T_1 = 0.1$, $T_2 = 1$, and $T_3 = 1.3$, where $T_2 = 1$ is the same as the target distributions introduced in App. A. We use the same architecture introduced in App. C.1, and the training of each model is long enough to ensure convergence. The ground truth of each benchmark at $T_1 = 0.1$ is obtained by running MCMC on top of the $T_2 = 1$ samples for a long enough time to ensure mixing.

Fig. 8 showcases the advantage of Finite-difference TE, where the extrapolated samples can be much closer to the ground-truth ones, compared to the other methods.

### D.2. Measuring Performance through Observables

Samples from the equilibrium (*i.e.* target) distribution are used for estimating the *observables* in many physical problems, which are mathematically the expectation of an *observable function* over the target distribution, *i.e.* $\langle O \rangle_X := \mathbb{E}_{p(x)}[O(x)]$. To measure the quality of generated samples by different models, we design a toy observable function, which is a quadratic

---

[4]DW-4 is a 4-particle system in 2-dimensional space introduced by Köhler et al. (2020). To clarify, we follow the notation used in (Köhler et al., 2020; Akhound-Sadegh et al., 2024) where the number 4 indicates the *number of particles*. While the number 2 in DW-2 indicates the potential is in a 2-dimensional space.

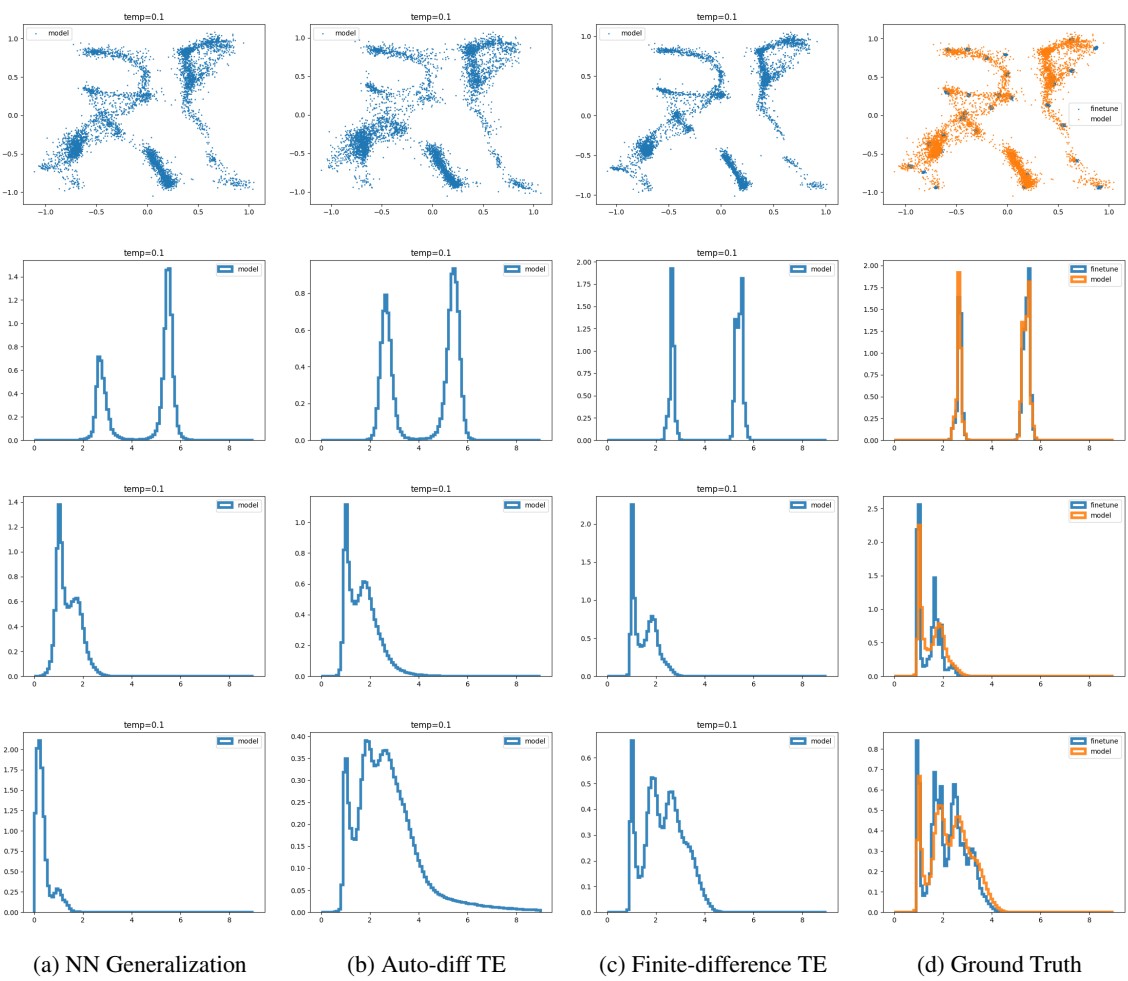

(a) NN Generalization      (b) Auto-diff TE      (c) Finite-difference TE      (d) Ground Truth

*Figure 8.* Different ways for extrapolation to temperature 0.1. In (d), oranges are Finite-Difference TE; blues are ground-truth, which can also be obtained by running 2 steps, 10 steps, 200 steps, and 200 steps of Langevin MCMC on top of Finite-Difference TE samples, respectively.

*Table 7.* **Comparing PTSD with other neural sampler baselines.** We measure a toy observable $O(X) = \mathbb{E}_{p(x)}[\lambda x^T x]$. We report the Mean Absolute Error (MAE) corresponding to the ground truth observable, which is estimated by 10000 samples from the target distribution. For each model, we use 10000 samples for estimating $O(X)$. '-' indicates that the method diverges or is significantly worse than others.

|  | GMM ($d = 2$) | MW32 ($d = 32$) | LJ55 ($d = 165$) |
|---|---|---|---|
| FAB | $\underline{1.91}_{\pm 0.98}$ | $4.09_{\pm 0.14}$ | - |
| CMCD | $7.03_{\pm 0.83}$ | $3.76_{\pm 0.12}$ | - |
| DDS | $6.85_{\pm 1.36}$ | $9.89_{\pm 0.13}$ | - |
| iDEM | $6.5_{\pm 0.75}$ | $9.28_{\pm 0.06}$ | $28.72_{\pm 0.02}$ |
| BNEM | $7.42_{\pm 0.63}$ | $11.09_{\pm 0.18}$ | $9.48_{\pm 0.01}$ |
| DiKL | $7.59_{\pm 0.76}$ | $\underline{3.24}_{\pm 0.17}$ | - |
| PT+DM | $12.09_{\pm 0.81}$ | $4.31_{\pm 0.10}$ | $\mathbf{0.34}_{\pm 0.03}$ |
| PTSD (ours) | $\mathbf{1.35}_{\pm 0.69}$ | $\mathbf{1.49}_{\pm 0.16}$ | $\underline{1.86}_{\pm 0.04}$ |

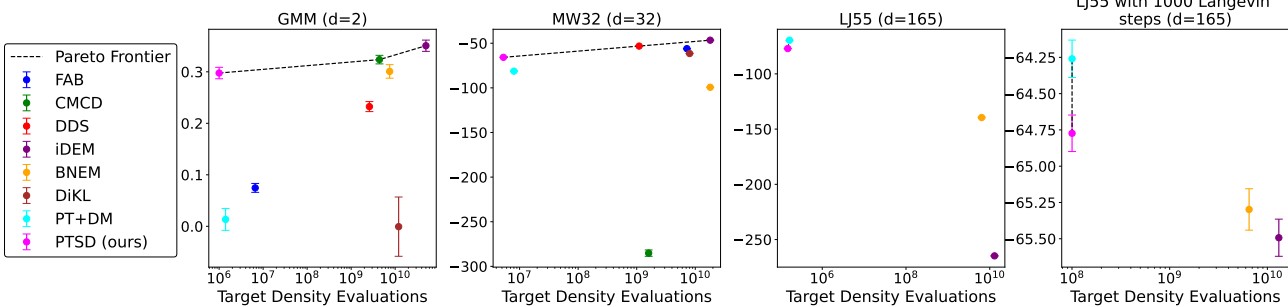

*Figure 9.* Log-likelihood $\mathbb{E}_{p(x)}[\log q_{model}(x)]$ and energy evaluations for all the models. To ensure consistency, the log-likelihood is calculated by fitting a diffusion model to samples generated from a given model.

function $O(x) = (x - a)^T C(x - a)$. To make the observable function be SE(3)-invariant for LJ system, we remove the mean of samples and ensure rotation-invariant for $O(x)$, *i.e.* $O(Rx) = O(x)$, by taking $a = 0$ and $C = I$.

Table 7 reports the MAE of the observable estimated by 10000 samples from different models, where the ground truth is obtained by Monte Carlo estimation through 10000 samples from the target distribution.

### D.3. Measuring Performance through Log-Likelihood

Negative Log-Likelihood (NLL) is a statistical metric that measures the distance between the target distribution $p$ and the probabilistic model $p_\theta$, which is computed as follows

$$\mathrm{NLL}(p_\theta; p) = -\mathbb{E}_p \log p_\theta(x). \tag{19}$$

Notice that computing NLL requires access to model density, which is intractable in both our method and most of baselines. Only FAB implemented with a Normalizing Flow has tractable model density. To remedy this, we train **an additional diffusion model** to 10000 samples from each model and evaluate the log model density of a generated sample $x_0$ through the *probability-flow ODE* (PF-ODE) as follows

$$\log p_\theta(x_0) = \log p_1(x_1) + \int_0^1 \nabla \cdot \tilde{f}(x_t, t)dt, \quad \text{with } x_t = x_1 - \int_1^t \tilde{f}(x_u, u)du. \tag{20}$$

Note that, while the above model density is exact and guaranteed by the instantaneous-change-of-variable, bias can be introduced in two ways: (1) the discretization; and (2) the variance of the estimation for the divergence term $\nabla \cdot \tilde{f}$ using Hutchinson's estimator.

The results are shown in Fig. 9. PTSD is consistently in the Pareto Frontier with respect to energy function evaluations and log-likelihood on all of the data sets.

### D.4. Alanine Dipeptide Experiment Setup and Results

We used 5 temperature geometrically spaced temperature levels for both the PTDM and PTSD models, spaced geometrically from 300K to 1500K. We initialised sampling by running a single PT chain on the top two temperatures for $1 \times 10^7$ steps (total $2 \times 10^7$ energy evaluations), and obtained a buffer of 200000 samples by subsampling. We trained the model for 100000 epochs on those 200000 samples with a learning rate of $2 \times 10^{-3}$. At each step, we then generate 100000 samples on the next level and do IS resampling with the truncation quantile set to 0.8. We pick a random subset of 1000 samples, and run PT for 1000 chains for each, and pick every 10th sample from these chains, resulting in 100000 new samples that we mix with the original IS resampled results. This requires $2 \times 1000 \times 1000 = 2 \times 10^6$ energy evaluations at each extrapolation step. We have 3 such steps, and the IS requires $100000 \times 3 = 3 \times 10^5$ energy evaluations, resulting in a total of $2 \times 10^7 + 2 \times 10^6 \times 3 + 3 \times 10^5 = 2.63 \times 10^7$ energy evaluations. We optimised the learning rate and truncation quantile hyperparameters to get to our final results. We chose the number 1000 for the amount of PT chains at each extrapolation such that we get a small enough energy evaluation budget, and early experiments showing that this was enough not to destabilize the training process.

For PTDM, we ran parallel tempering for 5.2e6 steps on the same 5 temperature levels, resulting in $5.2 \times 10^6 \times 5 = 2.6 \times 10^7$ energy evaluations. We trained the model for twice as long as we use to train each individual diffusion model in our PTSD implementation with a learning rate of $4 \times 10^{-3}$. We optimized over the learning rate to get to the final results. The setup of the 5 temperature levels was chosen after noticing in initial experiments that it works well for parallel tempering in particular, and we did not tune it for PTSD.

Fig. 10 shows the marginals of Fig. 7. Although PTDM does have less bias in the high-probability regions, the relatively short running time for PT has left it unable to effectively model the small mode on the right side of the $\phi$ plot. Table 3 shows the log-likelihoods as estimated with the probability flow ODE. We also evaluated the effective samples sizes (ESS), but the values for both diffusion models were very low (less than $1\%$) and showed high variance. We hypothesize that this is due to noise in the probability flow ODE causing some outliers with unusually low log-likelihood values, which causes these samples to dominate the importance weight distribution. In the experiments, we use truncated importance sampling to counter this. Fig. 11 shows the energy histogram of samples generated by the model, compared to the ground-truth distribution. The energies are in general higher for PTSD, indicating that the distribution is more spread out than the ground-truh distribution.

The reason we run MCMC from a subset of samples is that running it from all of the samples entangles the amount of samples we use for importance sampling with the amount of energy evaluations needed for MCMC. IS and MCMC have complimentary effects in the model: The benefit of importance sampling is that it is highly energy function evaluation efficient, using only one energy function per generated sample. The natural tradeoff is that it requires more compute for diffusion sampling and log-likelihood estimation. On the other hand, a benefit of MCMC sampling is that it can be used to increase the diversity of the IS outputs and obtain better coverage of the distribution for training the next diffusion model. Another advantage is that using correlated samples from the MCMC chains is also energy evaluation efficient. Accordingly, we noticed in our experiments that running a relatively small amount of longer MCMC chains from a large amount of IS generated samples results in a set of samples that has good distribution coverage while keeping the energy evaluation count low.

### D.5. Additional Visualizations for Generated Samples

Plots Fig. 12 and Fig. 13 visualise the GMM and MW32 samples from the different baseline models.

### D.6. Results on LJ55 with additional Langevin dynamics

The original evaluation scheme for the LJ55 task in (Akhound-Sadegh et al., 2024) involved running additional steps of Langevin dynamics on top of the pure neural sampler outputs. In our main table, we chose to follow (OuYang et al., 2024) and show the results for the pure neural sampler outputs, to more directly compare the performance of the base generative models. Here we provide results for the MCMC refined evaluation scheme, where we ran 1000 Langevin dynamics steps for each model generated sample. The sample evaluation results are shown in Table 8. The overall pattern is similar to the one in Table 1, where BNEM is the best, followed by PTDM, PTSD and iDEM. Fig. 9 shows the log-likelihood results with respect to energy function evaluations, placing PTSD and PTDM at the Pareto frontier.

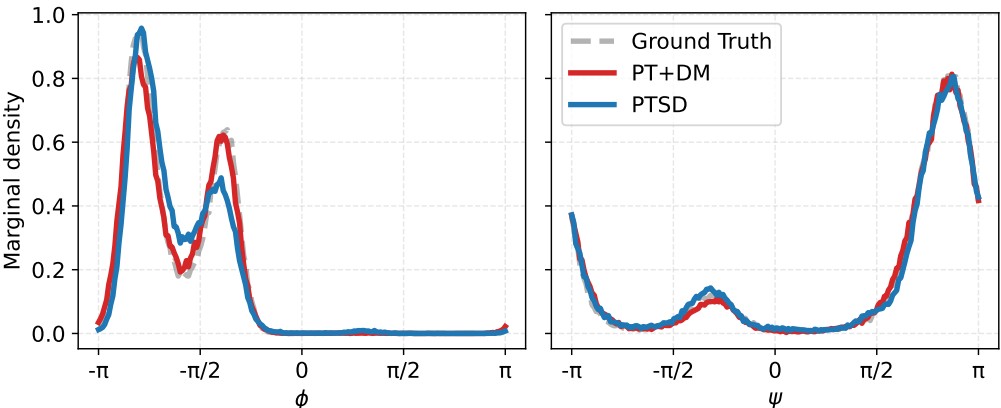

*Figure 10.* Marginal distributions of the $\phi$ and $\psi$ angles in the alanine dipeptide molecule. While PTDM captures the lower part of the $\phi$ values more accurately, the small mode on the right side of the $\phi$ plot is better represented by PTSD.

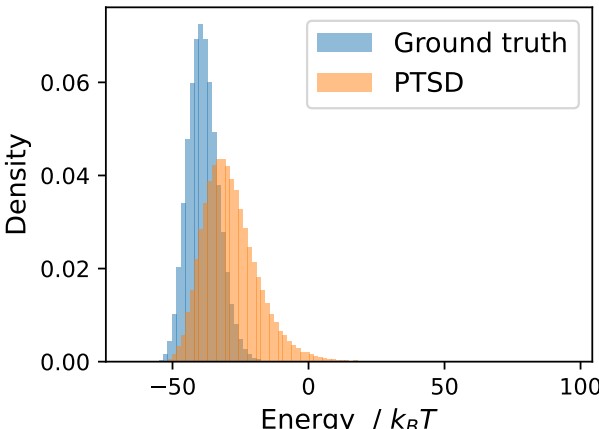

*Figure 11.* The energy histogram of samples generated by our model on the alanine dipeptide molecule, vs. the energy histogram of ground-truth samples. The PTSD samples tend have slightly higher energies, indicating that the distribution is slightly more spread out than the ground-truth distribution.

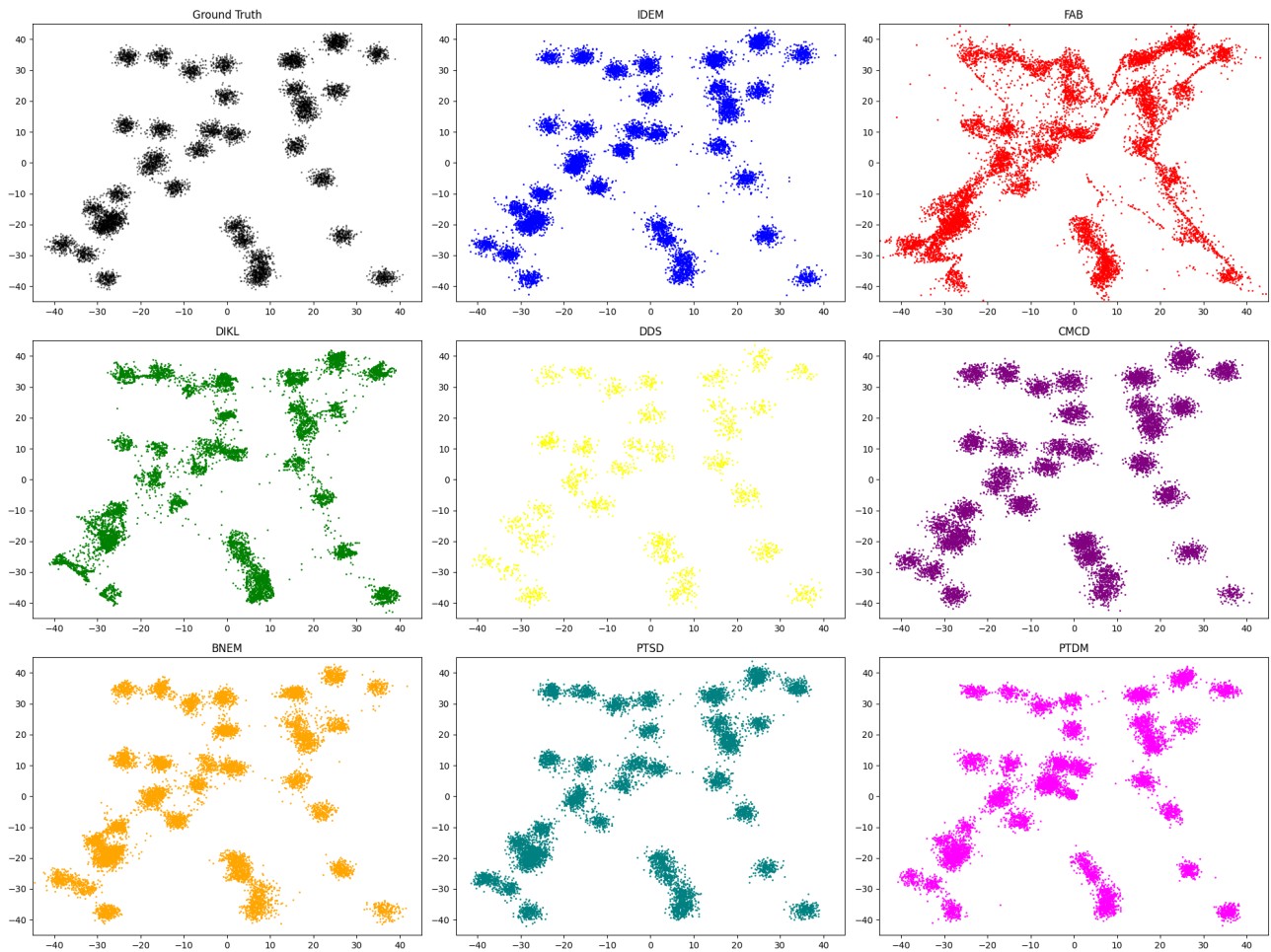

*Figure 12.* Visualising the samples generated by PTSD and other baselines on MoG-40.

*Table 8.* **LJ55 results for neural sampler methods using the evaluation scheme where we run additional MCMC steps on top of the generated samples.** We measure (**best**, second best) the TVD and MMD between Energy histograms, and $\mathcal{W}_2$ distance between data samples.

| | LJ55 ($d = 165$) | | |
| --- | --- | --- | --- |
| | TVD $\downarrow$ | MMD $\downarrow$ | W2 $\downarrow$ |
| iDEM | $0.42_{\pm 0.006}$ | $0.10_{\pm 0.001}$ | $1.83_{\pm 0.001}$ |
| BNEM | $\mathbf{0.09}_{\pm 0.004}$ | $\mathbf{0.00}_{\pm 0.000}$ | $\mathbf{1.81}_{\pm 0.000}$ |
| PT+DM | $\underline{0.15}_{\pm 0.004}$ | $\underline{0.01}_{\pm 0.001}$ | $\underline{1.82}_{\pm 0.001}$ |
| PTSD | $0.29_{\pm 0.001}$ | $0.05_{\pm 0.001}$ | $1.86_{\pm 0.001}$ |

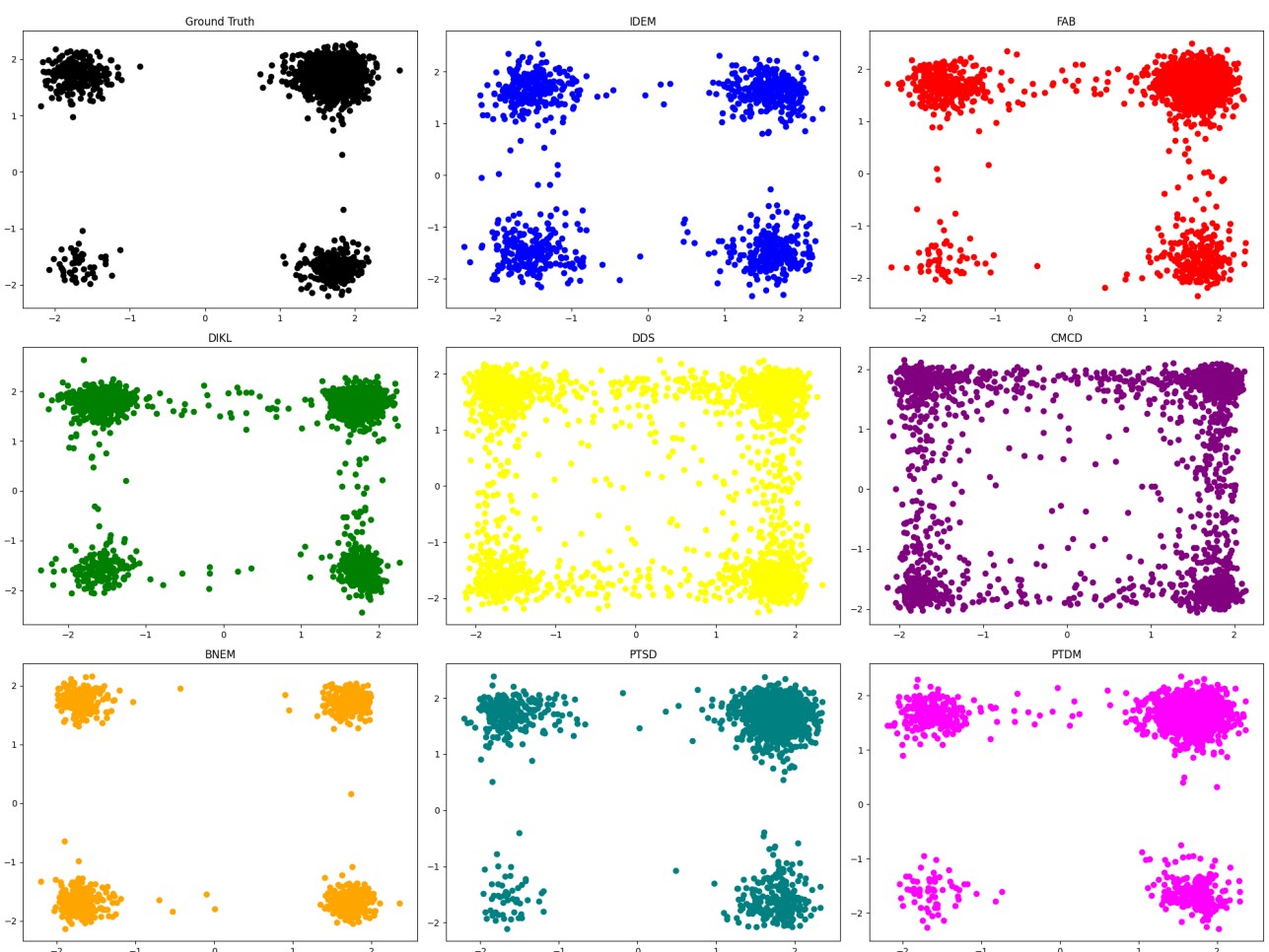

*Figure 13.* Visualising 2-dimensional slices of samples generated by PTSD and other baselines on MW-32.

