# OpenReview forum: "Progressive Tempering Sampler with Diffusion"
_ICML.cc/2025/Conference — ICML 2025 poster_

### Official Review · Reviewer_cnuL · 2025-03-06

**Overall Recommendation:** 1

**Summary:**

This paper proposes learning diffusion models at various temperatures using MCMC data at high temperatures. These models are then used to sample from unnormalized probability distributions. The authors employ a Taylor expansion of the target distribution over temperature to derive a temperature-dependent drift term, which is sequentially applied to generate samples at lower temperatures. Biases are corrected using importance resampling. The method is evaluated on three different target distributions commonly referenced in the literature.

**Claims And Evidence:**

The authors claim to achieve significant improvements in target density evaluation efficiency compared to previous diffusion-based neural samplers. However, upon examining Tables 1 and 2, I am not convinced that this claim is fully substantiated. To me it looks like PT+DM performs better in many cases and possibly insignificantly worse in some cases.

**Essential References Not Discussed:**

Essential references are discussed.

**Experimental Designs Or Analyses:**

The experimental design is logical, and some ablation studies are conducted.

**Methods And Evaluation Criteria:**

The evaluation criteria are sensible, but it would be beneficial if the authors also reported the Evidence Lower Bound (ELBO). The method is evaluated on only a few problem types, which is less comprehensive than typical evaluations in related literature. Additionally, the 2-D Gaussian Mixture Model (GMM) problem is quite simplistic.

**Other Comments Or Suggestions:**

- "PF-ODE" is inconsistently written in three different ways in the paper.
- Line 288: Consider providing a reference to the appendix and reporting the used truncation threshold.
- Lines 120 ff.: Although not part of the proposed method, could the authors elaborate on whether the swaps are performed with a certain probability after each MCMC update step?
- From where is the MW32 ground truth data obtained?
- L.127 ff. please make clear that you are using the Variance Exploding SDE. The notation with $\dot{\sigma}$ is rather unusual. so please explain that you are using the same notation as in Song 2021.

**Other Strengths And Weaknesses:**

**Strengths:**

- The idea of combining diffusion models with parallel tempering is compelling.

**Weaknesses:**

- Important experimental details are missing. For instance, for the LJ-55 problem, it is unclear what architectures are used for each method, how equivariance is ensured, and how many diffusion steps are employed.
- There is no documentation of the attempts made for methods that diverge on the LJ-55 problem.
- The experiments are limited to three tasks, one of which (GMM-2D) is overly simplistic. I suggest using more challenging alternatives like GMM-50D and Mixture of Students in 50D.
- The Taylor expansion with respect to temperature, where $\Delta T$ is significantly greater than 0, seems to be a crude approximation.
- The method introduces many additional hyperparameters (Tab. 4), and it is unclear how much tuning is required. It is also uncertain whether the same level of hyperparameter tuning was applied to other methods, particularly PT+DM.
- It is unclear what type of diffusion model was used for PT+DM. Overall, the experiments should be described in more detail.
- Figure 1 suggests that PTSD performs better than PT+DM, but Table 1 indicates that it is often outperformed by other methods.
- no error bars are reported

**Questions For Authors:**

- do you train your diffusion model using Hutchinson’s trace estimator or using score matching?
- How does your procedure compare to training CMCD and other diffusion samplers on data from parallel tempering and then fine-tuning it on the target distribution?
- How many samples are used for evaluation in Table 1?

**Relation To Broader Scientific Literature:**

The broader relation to scientific literature is established.

**Theoretical Claims:**

N/A

---

> ### Author Rebuttal · Authors · 2025-04-01
>
> Thank you for your detailed feedback and your constructive comments. We hope that our detailed responses would solve your concerns.Should you find our reply satisfactory, we kindly encourage you to raise your score.
>
> ## Q1: More details of the experiments
>
> Thank you for pointing out this. We will add them in our camera-ready version.  We now address your concern related to method details you mentioned.
>
>
> ### no documentation of the attempts made on the LJ-55 problem
> We did not save the results for LJ-55 for some baselines as it is significantly worse than others. A similar pattern has already been observed by iDEM (Fig 9).
> We note that these baselines will anyway require significantly more energy evaluation times than ours (and PT+DM).
>
> ### Details about additional hyperparameters in Tab 4
> For most of the targets, we tune PT+DM so that the swap rate is ~30%, and we use the same schedule for PT+DM and our approach. We also specifically tune PT+DM on GMM40 as it requires fewer levels. So we in fact put similar effort into tuning the temperatures for both approaches.
> However, we agree that our approach also has many other hyperparameters to tune, and this can make the tuning more involved. We will update our camera-ready paper to reflect this limitation.
> We also want to emphysize that more hyperparameters also means a **larger design space**, which has the potential to improve the performance of our approach further.
>
> ### More details about the diffusion model used for PT+DM
> To ensure fair comparison, we employ exactly the **same** neural network architecture and training hyperparameters for PT+DM and PTSD.
>
> ## Q2: The experiments are limited to three tasks
>  Following your suggestion, we evaluate our model on a more complicated task. As our main focus for the application is molecules, we evaluate our method on Alanine dipeptide (ALDP)  in Cartesian coordinates. This task is highly challenging for neural samplers.As we can see in https://anonymous.4open.science/r/PTSD-rebuttal-icml2025/ramachandran.png, our algorithm still obtain reasonable performance on this task, which strongly support the effectiveness of our approach.
>
> ## Q3: The Taylor expansion seems to be a crude approximation when $\Delta T$ not close to 0
> We totally agree. However, we found that this crude approximation provides a better exploration performance compared to using the derivative (with auto-diff) directly. We have provide a plot in https://anonymous.4open.science/r/PTSD-rebuttal-icml2025/extrapolation_ablation.png to illustrate this observation.
>
> This is possibly due to 2 reasons:
> 1. The derivative may not be robust, as we only train the model with a finite grid of temperatures;
> 2. Our approximation eventually forms a auto-guidance, which may have the potential to reduce the network imperfection.
> ## Q4: Figure 1 suggests that PTSD performs better than PT+DM, but Table 1 indicates that it is often outperformed by other methods
> We made a mistake in the previous LJ55 results. We now have updated the table in https://anonymous.4open.science/r/PTSD-rebuttal-icml2025/updated_main_table.png to reflect this. The energy function evaluation count decreased significantly, while also improving the energy-MMD. Note also the interatomic distance plot, which shows that our method beats BNEM in this regard.
>
> We can see that our approach delivers a performance that is better or on par with other baselines.
> We also note that even though sometimes the baselines outperform our approach, the improvement is very small. On the other hand, our method consistently delivers good results on these targets, and even works well on the challenging ALDP target.
>
> ## Q5: no error bars are reported
> Thank you for mentioning the missing or error bars, we will update them in our camera-ready version.
>
>
> ## Reply to Other Comments Or Suggestions
> Thank you for pointing out the inconsistency of the PF-ODE writing, the missing references, and clarity of the VE SDE writing. We will fix them. For the other questions:
>
> >How are swaps done?
>
> We use standard PT swap; the acceptance rate is in eq(2) in our paper.
>
> >How do we sample MW32 ground-truth?
>
> We follow FAB to first run iid samples from Double Well, and then stack 16 samples together.
>
>
> ## Responses to the Questions For Authors
> 1. We train our diffusion model by standard denoising score matching
>
> 2. For CMCD, and other diffusion samplers, it cannot directly train from data without increasing energy evaluation times. In fact, every time we calculate the loss, hundreds of energy evaluations. On the other hand, our pipeline does not need an extra energy evaluation and hence will be significantly more efficient compared to others.
>
> 3. We use 10000 samples

---

### Official Review · Reviewer_WWs4 · 2025-03-13

**Overall Recommendation:** 1

**Summary:**

The paper introduces a new sampling algorithm—Progressive Tempering Sampler with Diffusion (PTSD)—which aims to efficiently sample from unnormalized densities by combining ideas from traditional parallel tempering (PT) and modern diffusion-based neural samplers. The method is evaluated on several multimodal targets, where it demonstrates significant improvements in sample quality and target density evaluation efficiency over existing neural samplers.

**Claims And Evidence:**

Yes the claims look clear and are supported by basic math and some limited experiments.

**Essential References Not Discussed:**

Everything is cited.

**Experimental Designs Or Analyses:**

Yes they look valid.

**Methods And Evaluation Criteria:**

Yes they do.

**Other Comments Or Suggestions:**

No other comments.

**Other Strengths And Weaknesses:**

Weaknesses:
* the paper is hard to read in general and is applied only to a niche audience.
* The proposed PTSD method is kind of slow in the experiments conducted. Any specific reason for that?
* I would also say that the method is very limited to simple models. The authors have to try it to bigger and more state of the art models.
* The experiments in general seem kind of simplistic and the results in Table 1 are also not that good. Is there a reason for that? Cause in its current form the paper is not strong either mathematically or experimentally.

**Questions For Authors:**

Please check the Weaknesses.

**Relation To Broader Scientific Literature:**

I would say the findings are kind of limited and not very interesting to the broader scientific audience.

**Theoretical Claims:**

Yes I checked the math but the paper lacks proofs.

---

> ### Author Rebuttal · Authors · 2025-04-01
>
> Thank you for your review. We hope that our detailed responses would solve your concerns. If any questions or concerns arise, please do not hesitate to let us know, and we will address them properly. However, should you find our reply satisfactory, we kindly encourage you to raise your score.
>
>
> ## Q1: the paper is hard to read in general and is applied only to a niche audience
> Thank you for your comment. We will be happy to update our paper if you could kindly provide more information in detail about the part that you found hard to read.
> Additionally,we respectfully disagree with the argument that this paper is niche. Sampling from unnormalized densities is a long-standing task in many areas, including Bayesian inference, Molecular simulation, physics, statistical chemistry, etc. There are many approaches trying to address this problem. Here is a non-exhaustive list of recent advancements:  [1, 2, 3, 4, 5].
>
> ## Q2: The proposed PTSD method is kind of slow in the experiments conducted. Any specific reason for that?
> This seems to be a misunderstanding. As we can see from Fig 6 and Tab 2 in our manuscript, our method is actually the most efficient approach in terms of energy function evaluations. It is order-of-magnitude faster than other neural sampler baselines. Could you specify where you found our method slow?
>
> ## Q3: Proposed method is limited to simple models
> Our method does not rely on any specific architecture. One can apply any architecture to our proposed pipeline.
>
> Also, for the problem we are addressing, we believe the model (e.g., EGNN) we use is indeed “SOTA” as it balances well running cost and performance. The architecture we use is a standard choice for the n-body system and small molecules [4, 6, 7]. Therefore, it would be very helpful if you could clarify what kind of model you expect us to evaluate our approach on.
>
> ## Q4: The experiments in general seem kind of simplistic, and the results in Table 1 are also not that good. Is there a reason for that?
>
> Thank you for your comment. We used the wrong hyperparameter (model size) when we ran the results for Table 1, and hence LJ55 is slightly worse than expected. We have fixed this issue, and we can see the updated results in https://anonymous.4open.science/r/PTSD-rebuttal-icml2025/updated_main_table.png, where both the energy-MMD and data-W2 are improved. We also see that number  of energy function evaluations of our approach is **less** than PT+DM across all 3 tasks. The visualization of the interatomic distance in https://anonymous.4open.science/r/PTSD-rebuttal-icml2025/LJ55_interatomic.png also reveals that our method does have advantages over BNEM in this regard, and is a strong competitor to PT+DM.
>
>
> Additionally, we provide results on **alanine dipeptide** in Cartesian coordinates, which is a challenging task where most of the baselines failed. This significantly highlights the capability of our approach. The experimental results can be found in https://anonymous.4open.science/r/PTSD-rebuttal-icml2025/ramachandran.png.
>
>
> ## Q5: Paper lacks proof
> Our approach relies on Taylor expansion and finite difference approximation for first-order derivatives. There are standard results in calculus.
> Could you please explain which part you suggest us to include a proof?
>
>
> [1] Noé, Frank, et al. "Boltzmann generators: Sampling equilibrium states of many-body systems with deep learning." Science 2019.
>
> [2] Doucet, Arnaud, et al. "Score-based diffusion meets annealed importance sampling." NeurIPS 2022.
>
> [3] Midgley, Laurence Illing, et al. "Flow Annealed Importance Sampling Bootstrap." ICLR 2023.
>
> [4] Akhound-Sadegh, Tara, et al. "Iterated denoising energy matching for sampling from boltzmann densities." ICML 2024.
>
> [5] Vargas, Francisco, et al. "Transport meets Variational Inference: Controlled Monte Carlo Diffusions." ICLR 2024.
>
> [6] Hoogeboom, Emiel, et al. "Equivariant diffusion for molecule generation in 3d."  ICML 2022.
>
> [7] Klein, Leon, Andreas Krämer, and Frank Noé. "Equivariant flow matching." NeurIPS 2023.

---

### Official Review · Reviewer_v828 · 2025-03-13

**Overall Recommendation:** 3

**Summary:**

This paper focuses on the problem of sampling from the unnormalized densities. The authors recognize the drawbacks of MCMC with Parallel Tempering as well as the ones of recent neural samplers (mostly based on the diffusion process). In particular, they proposes a novel sampling method  — Progressive Tempering Sampler with Diffusion — that trains diffusion models sequentially across temperatures. The main idea is to start from training the high-temperature diffusion models, which are used to generate a lower-temperature samples, slightly refined with MCMC approach. Such samples are then used to train the lower-temperature diffusion model. The method is compared against other sampler on standard benchmarks, where the authors focuses on the number of target evaluations during training and sample-based metrics.

**Claims And Evidence:**

The claims regarding findings the advantages and drawbacks of neural samplers and Parallel Tempering, as well as introducing a novel method (being more efficient in terms of number of target evaluations during training) are well justified. Same, regarding the theoretical findings.

However, I think that the claimed superiority to neural samplers in terms of generated samples and better scalability is not well justified, mostly due to the limited empirical evidence.

**Essential References Not Discussed:**

The references are discussed in general.

**Experimental Designs Or Analyses:**

Yes, for part of my concerns, please see the Methods And Evaluation Criteria section. Moreover, I generally think that the samples quality evaluation is missing, e.g., plot of interatomic distances for LJ potentials (comparing other methods), logZ or NLL metric, etc.

**Ablation studies:** I think that the influence of used sequences of temperatures during training, and the influence of a buffer size might be beneficial for this work. Moreover, I would like to veto see the comparison between models on their memory costs.

**Methods And Evaluation Criteria:**

Yes, but they are limited. I would like to see the comparison against other neural samplers like PIS, DIS, or GFlowNets. Moreover, I’m concerned by the lack of standard metrics for sampling problems (logZ, NLL, etc.), which are missed in the experiments.

In particular, the authors do not provide the results for FAB on LJ55, which was one of the first methods checked in this setting and the results for TVD for iDEM are various from the ones from the original paper.

**Other Comments Or Suggestions:**

For comments and suggestions, please see the previous sections.

**Other Strengths And Weaknesses:**

**Strengths:**

[1] Proposing a novel method of combining the PT with diffusion for more efficient training of samplers.

[2] The method seems to be theoretical justified, and significantly lowered the number of needed target energy evaluation.


**Weaknesses:**

[1] Missing results for some benchmarks (like FAB).

[2] Missing significant metrics for sampling quality.

[3] Lack of comparison of memory cost of training samplers, and lack of scaling PTSD into more complex problems (since it requires lower number of target energy evaluations).

[4] Regarding all of the mentioned weaknesses, I think that this paper (in the current form) has limited significance.

**Questions For Authors:**

For questions, please see the previous sections.

**Relation To Broader Scientific Literature:**

The key findings are based on existing knowledge and methods (PT and diffusion), but the proposed method is novel.

**Theoretical Claims:**

Yes, I've checked and haven't found any obvious drawbacks.

---

> ### Author Rebuttal · Authors · 2025-04-01
>
> Thank you for your detailed feedback and your constructive comments. We now reply to your concerns one by one. Should you find our reply satisfactory, we kindly encourage you to raise your score.
>
> ## Q1: Limited empirical evidence & lack of scaling PTSD into more complex problems
>
> To showcase the superiority and better scalability of our method, we conduct experiments on a more complex system-Alanine Dipeptide in Cartesian coordinates, a task that most of the neural samplers failed on. The results of experiments are provided in https://anonymous.4open.science/r/PTSD-rebuttal-icml2025/ramachandran.png. As we can see, our algorithm provides high-quality samples on this challenging task, with very few energy function evaluations. (uses almost an order of magnitude less evals than FAB)
>
> We will include this results in our camera-ready version.
>
> ## Q2: lack of comparison against PIS, DIS, or GFlowNets
> Thank you for suggesting these baselines. However, we highlight that PIS, DIS and GFN essentially have the same property as DDS: they mostly use the same model architecture as DDS (parameter the network with a score term); and they require simulating the entire trajectory, which requires hundreds of energy calls to calculate the loss once. Therefore, we use DDS as a representation for this family of approaches.
>
> ## Q3: Lack of results for FAB on LJ55
>
> The result for FAB on LJ55 is not shown because its training diverged in our experiments. Similar results were also observed in iDEM: From iDEM Fig 9, we can see FBA is significantly worse than iDEM.
>
>
>
> ## Q4: Different results for iDEM on LJ55
> We note that in iDEM, they obtain their results on LJ-55 by running a few steps of Langevin dynamics on top of the generated samples. This can be seen from their codes:
>  https://github.com/jarridrb/DEM/blob/main/configs/experiment/lj55_idem.yaml where they set “num_negative_time_steps: 10”. This means they run 10 steps of Langevin dynamics in https://github.com/jarridrb/DEM/blob/main/dem/models/components/sde_integration.py
>
> As this trick is applicable to our approach and all the baselines, we present the results without it.
>
> ## Q5: lack of logZ, NLL, etc.
> Computing logZ and NLL requires model density. Different baselines will have different ways to calculate logZ and NLL: For CMCD and DDS, we need to discretize the SDE into sequence of Gaussians; for diffusion models (ours and PT+DM), we can either use this approach, or estimate it with change of variable on PF-ODE. As different approaches have different discretizations or different underlying processes, this would make a fair comparison more difficult.
>
> In the end of the day, one cares ultimately about the quality of the samples. Therefore, we believe that sample-based metrics are more straightforward and comparable across methods.
>
> ## Q5: Lack of interatomic distances for LJ
> we have provided it in https://anonymous.4open.science/r/PTSD-rebuttal-icml2025/LJ55_interatomic.png. We see that PT+DM is a strong baseline for LJ55, but PTSD is a close second in terms of visually matching with the ground-truth distribution.
> We will add this visualization in our camera-ready version for a more comprehensive evaluation.
>
>
> ## Q6: Ablation studies
> Following your suggestion, we studied the effect of the amount of temperatures in the GMM task, and report the results in https://anonymous.4open.science/r/PTSD-rebuttal-icml2025/num_temp_ablation.png.
>
> We can see the trend is monotonic: all of the metrics improve with more temperature levels. This indicates we can always use more compute to obtain better results in our method.
>
>
> ## Q7: Lack of comparison of the memory cost of training samplers
> Our approach encounters a larger memory cost in two places. We now analyze these two costs one by one:
>
> 1. When we run MCMC on the samples from the buffer, if we parallelize them, we will encounter a larger memory cost compared to PT.
> In fact, we view this as an advantage rather than a limitation as this indicates that our method enables efficient parallelization of MCMC into multiple independent chains, with the number of parallel chains determined by the buffer size. On the contrary, standard PT can only parallelize over as many chains as the number of temperatures. The buffer size is normally much larger than the number of temperatures; our approach can, in fact, allow for more massive parallelization.
>
>
> 2. When we sample using our proposed temperature-guided score, we will need to save and call two models together in memory; we may also need to backprop through these two models. In total, they will increase the memory cost to ~4x of standard diffusion model sampling.
> We agree that this a potential limitation for our approach, and we will include a discussion on this in our camera-ready version.
>
> Thank you for this insightful question. This allows us to realize an additional advantage and limitation of our algorithm, which we believe have made our paper more solid and stronger.

---

> > ### Comment · Reviewer_v828 · 2025-04-04
> >
> > Thanks for the rebuttal and authors' work. Some of my concerns were addressed, but not all of them. I believe that the comparison against iDEM, not including their approach to use Langevin steps on top is a little unfair. Because of that, the results from iDEM differ much than those from this manuscript.
> >
> >
> > I think some way to comparing the NLLs is needed, because we know that pure sample-based metrics (e.g., $W_2^2$) behaves differently than log-prob like measures. For example, finding the means of the modes is enough in many cases to obtain a low value of Wasserstein distance. I do not agree that we only care about the sample quality and that the sample-based metrics are enough to evaluate them.
> >
> >
> > Finally, the presented Ramachandran plots for PTSD look worse than the ones from FAB original paper (Fig. 4), so I believe more extensive evaluation is needed.
> >
> >
> > Once again, I thank the authors for their work, but I think the current version of the evaluation setting (e.g., irreproducibility of iDEM/FAB results, missing computational cost comparison) is not enough. I will keep my score.

---

> > > ### Author Response · Authors · 2025-04-08
> > >
> > > Thank you very much for the response and the further constructive suggestions. Below, we try to address your remaining concerns:
> > >
> > > ## 1. LJ55 results with Langevin steps:
> > >
> > > Following your suggestions, we provide an extra comparison against iDEM that includes their approach of using Langevin steps.
> > > Specifically,  we rerun the LJ55 evaluation by using 1000 Langevin dynamics steps on top of the samples for all of the methods. We include the interatomic distance histograms for the methods with Langevin dynamics in https://anonymous.4open.science/r/PTSD-rebuttal-icml2025/LJ55_with_Langevin_interatomic.png. Below, we also list the TVD/MMD/W2 metrics with this evaluation method:
> > >
> > >
> > > | Method | TVD | MMD | W2 |
> > > |--------|-----|-----|-----|
> > > | iDEM | 0.41 | 0.11 | 15.8 |
> > > | BNEM | 0.07 | 0.003 | 15.7 |
> > > | PTSD | 0.28 | 0.05 | 16.8 |
> > > | PTDM | 0.14 | 0.017 | 15.8 |
> > >
> > > The overall comparative pattern remains the same. We will also include uncertainty intervals in the camera-ready version. However, we still note that comparing the methods without Langevin steps is the fairest way for three reasons:
> > > 1. These Langevin steps are not part of iDEM’s approach.
> > > 2. Langevin dynamics can be applied to all of the samplers efficiently.
> > > 3. Langevin steps can reduce the inherent differences between different samplers: the better the sampler is, the less improvement it can gain by running Langevin dynamics.
> > >
> > > Therefore, we will include both results (w and w/o Langevin) in our camera-ready version.
> > >
> > > ## 2. NLL
> > > Following your suggestions, we now provide NLL for our approach and baselines. To evaluate NLLs in a fair way, we used the same approach as iDEM by training diffusion models on samples generated by the different models, and defining the likelihoods using the resulting diffusion probability flow ODE, estimated using Hutchinson’s trace estimation trick and 2000 Euler integration steps. We include the results in https://anonymous.4open.science/r/PTSD-rebuttal-icml2025/nll.png. We can see that PTSD is always on the Pareto frontier of energy function evaluations and data log-likelihoods among the compared methods, often with large margins.
> > >
> > > ## 3. Differences between our alanine dipeptide results vs. FAB
> > > For the ALDP experiment, we would highlight three significant differences between our method and FAB:
> > > 1. In our experiment, the results are produced by using only $2.6\times 10^7$ energy-function-evaluations, while FAB uses $2.0 \times 10^8$, which is almost an order of magnitude more expensive (see Table 8 in [1]). As such, we think that these two models should be interpreted as targeting different energy evaluation ranges.
> > > 2. Our method samples in Cartesian coordinates, while FAB samples ALDP in Internal coordinate. Cartesian coordinates significantly complicate the task.
> > > 3. The FAB results are presented after importance resampling, whereas our samples are direct samples from the model. We updated the Ramachandran plots in https://anonymous.4open.science/r/PTSD-rebuttal-icml2025/ramachandran.png, where the result of FAB is obtained from (Figure 4.) in [1].
> > >
> > > We agree that a more detailed analysis of the results is useful, and will include more detailed comparisons like NLL in the updated manuscript.
> > >
> > > Thank you again for your valuable and constructive review. We hope that our responses have effectively addressed your remaining concerns, and hope that you could increase the rating accordingly if we have addressed your concerns successfully.
> > >
> > > [1] Midgley, Laurence Illing, et al. "Flow Annealed Importance Sampling Bootstrap." ICLR 2023.

---

### Official Review · Reviewer_G8RJ · 2025-03-14

**Overall Recommendation:** 4

**Summary:**

The article describes a method based on diffusion model in order to sample from unnormalized densities, such as $p(x) P \frac{\tilde{p}(x)}{Z}$. The approach is based on parallel tempering: many diffusion models are trained to match the distribution $p$ at various temperature. The general idea being that at large temperature, it is easy both to produce samples from $p$ and to reproduce faithfully the distribution's density using the samples. Therefore, first models are trained at large temperature: samples are easily generated using MCMC and the diffusion model are trained upon them. Then, lower temperature diffusion model are trained using eq. (11) starting from the weights learned with a previously trained machine (at a slightly higher temperature). The samples are corrected using importance sampling to correct the bias.

**Claims And Evidence:**

The paper does not claim any particular theoretical or practical results. It is mostly based on an expansion of the score function as a function of a temperature parameter in order to learn diffusion model for the unnormalized density at various temperature. It shows that this approach seems to work in a set of benchmarks.

**Essential References Not Discussed:**

I did not find that an important reference was missing.

**Experimental Designs Or Analyses:**

The experimental design is sound.

**Methods And Evaluation Criteria:**

The method is based on the training of various diffusion models to adjust on a probability density that is annealed from a "high temperature" regime where it is easy to sample from and from which samples are first generated with Monte Carlo to train the first stage of the diffusion models. Then, using both a Taylor expansion of the score function and importance sampling, it progressively learn the set of diffusion models to adjust toward the target distribution reaching $T=1$ for the annealing parameters, without the need to sample from the target distribution.
The method is evaluated on a set of benchmarks and compare to other methods. Three different measures are used to compare the different models.

Overall the criteria seems adequate for the considered setting.

There is no comment about how hard these benchmarks are in the text, such as a measured Monte Carlo relaxation time or any quantitative criteria.

**Other Comments Or Suggestions:**

no additional suggestions.

**Other Strengths And Weaknesses:**

- The proposed method combining PT and diffusion models is original, interesting and promising.
- The main limitation might be the burden to train a large amount of diffusion models, when the number of temperatures levels become large
the computational cost  and how this compares to other methods is not clearly discussed

**Questions For Authors:**

This work shows how to use tempering to learn unnormalized probability distribution. I have some questions:
* the authors use the method for instance on Lennard-Jones particular. Is there any metric related to the physics of the probability distribution that can be used in such case to attest the "goodness" of the generated samples ?
* How does the method depend on the number of temperatures ?
* The method seems to have a better W2 distance in general than the other methods, is there a reason why it should be the case ? (the other indicators are not as different between the different methods).
* I have the impression that one weakness of the method is the lack of "proof" that the algorithm should converge toward the correct probability distribution in some limit. For instance, with MCMC, in theory we can always perform a good sampling by increasing the number of MC steps. Would there be some similar results for the setting used in this work ?

**Relation To Broader Scientific Literature:**

The method is related to parallel tempering. An aspect which limits the use of PT is the presence of first order transition along the temperature path, which typically occurs when the data distribution has well separated modes which is quite common in ML. This requires then extensive number of temperatures. This is discussed for instance in Béreux et al. "Fast, accurate training and sampling of Restricted Boltzmann Machines"  (ICLR 2025). I would imagine that the temperature expansion which is used in the diffusion model is going to break down as well, rendering the method inoperative in this case?

**Theoretical Claims:**

There are no theoretical claims.

---

> ### Author Rebuttal · Authors · 2025-04-01
>
> Thank you for your detailed feedback and your constructive comments. We hope that our detailed responses would solve your concerns.
>
> # Questions:
> ## Q1: Metrics related to physics
>
> We follow the metrics used in previous literatures, e.g., iDEM, NEM, DiKL, FAB, etc. In these baselines, they do not evaluate on any physical metrics.
>
> One potential metric to use is to estimate the expectation of some manually crafted function using samples generated by our approach, and compare it with the value estimated using ground truth samples. We can view this as a “toy”  physical property. In practice, we use a quadratic function of the configuration, i.e., estimate $\mathbb E_{p(x)}[ x^T x ]$. We compare the estimated integral with the ground-truth using the Mean Absolute Error (MAE) and normalise the error as a percentage of the ground-truth integral value. We estimate 90% confidence intervals with bootstrap resampling. As we can see, our method can still deliver good results consistently.
> We will include this result in the camera-ready version.
>
> | Method | MW32         | LJ55           |
> |--------|--------------|----------------|
> | IDEM   | 8.97 ± 0.61  | 28.7251 ± 0.02 |
> | FAB    | 4.08 ± 0.14  | -              |
> | DIKL   | 3.19 ± 0.26  | -              |
> | DDS    | 9.89 ± 0.12  | -              |
> | CMCD   | 6.71 ± 0.26  | -              |
> | BNEM   | 10.61 ± 0.27 | 11.61 ± 0.05   |
> | PTSD   | 1.49 ± 0.16  | 1.87 ± 0.07   |
> | PTDM   | 4.32 ± 0.16  | 0.34 ± 0.04            |
>
>
> ## Q2: How does the method depend on the number of temperatures?
>
> More temperatures can (1) reduce the burden of training models for each temperature, and (2) can make our temperature guidance approximation more accurate.
> (1) is because, in practice, we always **fine-tune** our new diffusion model from the model trained on the last temperature. Using more temperature levels can reduce the difference between levels.
> For (2), we can look at Eq (11) in our manuscript to see this. When T1 -> T2, the guided score will become accurate. However, similar to PT, running our approach with too many temperatures can make the pipeline less efficient. In practice, we found matching the temperature schedule with that of standard PT can usually lead to a good performance. We provide an additional ablation study for it, by differing number of temperatures for training PTSD on the GMM problem: https://anonymous.4open.science/r/PTSD-rebuttal-icml2025/num_temp_ablation.png.
>
> ## Q3: Why is the W2 distance better, but the other metrics are not as different between different methods?
>
> This is an interesting observation. We suspect that this is because W2 directly measures the difference between sets of samples, while TVD and MMD are computed  on the energy values for the samples, which loses more information.
> We can also see our approach achieves better results in the above, reflecting a consistent trend with W2.
>
>
> ## Q4: Convergence of the proposed method:
> > one weakness of the method is the lack of "proof" that the algorithm should converge toward the correct probability distribution in some limit.
>
> In fact, we can always get better results by either increasing the number of temperatures or increasing the local parallel tempering refinement steps.
> We can view these limits as just running annealed Langevin with a model trained along the sampling procedure. In this case, the only error will be the intrinsic model imperfection. Therefore, the convergence is in fact guaranteed.
>
> # Other concerns:
> ## Q5: Potential failure case for target distribution with well-separated modes
> Thank you for raising this insightful concern. We agree that our algorithm could potentially require many temperature levels for good performance if standard PT suffers due to some phase transition / separated modes. However, we emphasize that this limitation is not unique to our approach -  standard PT can also require more chains and require task-specific design (for example, the trajectory PT in the case you mentioned).
> We will include this discussion in our camera-ready version, and we believe this discussion could make our paper stronger.
>
> ## Q6: Computational cost
> >The main limitation might be the burden to train a large amount of diffusion models, when the number of temperatures levels become large [...]
>
> We agree that this can be slow. However, in practice, we always **fine-tune** our new diffusion model from the model trained on the last temperature. Using more temperature levels can reduce the difference between levels, and hence, the training cost per level will also be reduced.

---

### Decision · Program_Chairs · 2025-05-01

**Decision:**

Accept (poster)

**Comment:**

This paper proposes a method for training diffusion samplers of unnormalised densities by iteratively training diffusion models on buffers of samples obtained by parallel tempering (PT) at decreasing temperatures. Notably, this is a simulation-free algorithm, as it involves only data-based (score matching) training of diffusion models, unlike the most successful prior methods; however, it is asymptotically unbiased only in the limit of converged PT at each iteration.

There was disagreement among the reviewers about the fairness of the comparisons (to iDEM with/without Langevin postprocessing and to methods that diverged when run out of the box). The responses largely addressed these concerns, and the authors are expected to include the new results in the final version.

Note: Reviewer cnuL initially did not respond to the authors' rebuttal and AC's requests for comment. While some of their comments have been substantially answered, in subsequent discussion the reviewer was still concerned about reproducibility / experiment details and rigour, and the authors are asked to take this feedback into account (reporting all experiment details and giving error estimates as promised). The review by Reviewer WWs4 was discounted, as it made a number of inaccurate comments, while others were answered in the response.